# Spatial distribution of water level impact to back-barrier bays

Alfredo L. Aretxabaleta[1], Neil K. Ganju[1], Zafer Defne[1], and Richard P. Signell[1]

[1] U.S. Geological Survey, Woods Hole, Massachusetts, 02543, USA

*Correspondence to*: Alfredo L. Aretxabaleta (aaretxabaleta@usgs.gov)

**Abstract.** Water level in semi-enclosed bays, landward of barrier islands, is mainly driven by offshore sea level fluctuations that are modulated by bay geometry and bathymetry, causing spatial variability in the ensuing response (transfer). Local wind setup can have a complementary role that depends on wind speed, fetch, and relative orientation of the wind direction

and the bay. Bay area and inlet geometry and bathymetry primarily regulate the magnitude of the transfer between open ocean and bay. Tides and short-period offshore oscillations are more damped in the bays than longer-lasting offshore fluctuations, such as storm surge and sea level rise. We compare observed and modeled water levels at stations in a mid-Atlantic bay (Barnegat Bay) with offshore water level proxies. Observed water levels in Barnegat Bay are compared and combined with model results from the Coupled Ocean-Atmosphere-Wave-Sediment Transport (COAWST) modeling system

to evaluate the spatial structure of the water level transfer. Analytical models based on the dimensional characteristics of the bay are used to combine the observed data and the numerical model results in a physically consistent approach. Model water level transfers match observed values at locations inside the Bay in the storm frequency band (transfers ranging from 50-100%) and tidal frequencies (10-55%). The contribution of frequency-dependent local setup caused by wind acting along the bay is also considered. The wind setup effect can be comparable in magnitude to the offshore transfer forcing during intense

storms. The approach provides transfer estimates for locations inside the Bay where observations were not available resulting in a complete spatial characterization. An extension of the methodology that takes advantage of the ADCIRC tidal database for the east coast of the United States allows for the expansion of the approach to other bay systems. Detailed spatial estimates of water level transfer can inform decisions on inlet management and contribute to the assessment of current and future flooding hazard in back-barrier bays and along mainland shorelines.

## 1 Introduction

Back-barrier bays or coastal lagoons are common features along the coast of the United States. Their depths are usually on the order of a few meters and their horizontal extents are on the order of several tens of kilometers. They are often surrounded by highly populated areas and susceptible to intense human and environmental stressors. During storms, surge

and larger than normal waves combine to inundate low-elevation areas, resulting in hazards to coastal communities. Both

hurricanes and winter storms affect coastal populations, infrastructure, and natural resources along the coastal bays of the United States (Nicholls et al., 2007, 2014; Rahmstorf, 2017; Wahl et al., 2017).

Hazard assessments consist of a characterization of the spatial and temporal extent of damaging physical events and the determination of the specific characteristics of those events (Ludwig et al., 2018). While flooding in the mainland side of back-barrier bays has severe socio-economic implications, most of the coastal hazard evaluations (Gornitz et al., 1994; Thieler and Hammar-Klose, 1999; Klein and Nicholls, 1999; Kunreuther et al., 2000; Neumann et al., 2015; Vitousek et al., 2017) have focused in open-coast areas. Vulnerability evaluation of coastal areas around back-barrier bays requires extensive knowledge of the main hazard sources and their physical controls.

Water level in the bays is mainly driven by offshore sea level fluctuations with additional effects by local wind and wave setups. The bay exchange with the ocean usually occurs through narrow inlets. The size of the inlet determines the frictional effects and the amount of dampening offshore fluctuations encounter (Keulegan, 1967). Tides and short-period offshore oscillations tend to be more dampened in the bays than longer-lasting offshore fluctuations, such as storm surge and sea level rise.

Bay water level fluctuations are linked to offshore forcing especially at low frequencies, while wind acting directly over the bay is more connected to current fluctuations in the bay (Garvine, 1985). Chuang and Swenson (1981) determined that water level changes at subtidal frequencies in Lake Pontchartrain were controlled by coupled coastal ocean-bay fluctuations. Wong and Wilson (1984) studied subtidal sea level fluctuations in Great South Bay and again found them primarily driven by bay-shelf coupling. In Delaware Bay, a bay-inlet system with a relatively large opening, Wong and DiLorenzo (1988) showed that remote effects dominate over local effects and that fluctuations at both tidal and subtidal frequencies in connected bays of the Delaware Bay system were forced by shelf sea level.

More recently, Aretxabaleta et al. (2014) analyzed water level data in Barnegat Bay and Great South Bay before and after Hurricane Sandy and demonstrated that the offshore-bay transfer was not significantly altered by the geomorphologic changes caused by the storm. Aretxabaleta et al. (2017) described observed changes in both tidal amplitude and bay water level transfer from offshore in Great South Bay and connected bays and related the changes to the dredging of nearby inlets and the changing size of a breach across Fire Island caused by Hurricane Sandy. They also introduced an analytical model, based on the Chuang and Swenson (1981) approach but extended to interconnected bays, that incorporated bay and inlet dimensions and matched the observed transfer of offshore water level fluctuations into the bay system.

In this study, we combine an analysis of observed water levels in Barnegat Bay with the results of numerical models and an analytical description of the system to characterize the spatial characteristics of the bay response to offshore fluctuations. The observations provide detailed information at five locations in the bay, while the numerical simulations can expand the

analysis to the entire bay system. The analytical model allows for the evaluation of the importance of the dominant factors affecting water level in bays. The combined approach can be used to provide consistent spatial maps of offshore water level impact into back-barrier bays. The method will be useful for coastal hazard assessment assisting in the management of nuisance flooding (Moftakhari et al., 2018) and providing spatial differences in vulnerability to perigean spring tides (king tides) and planning for flooding in response to storms of different durations. The final hazard estimates will be included as part of the U.S. Geological Survey Coastal Change Hazard portal (USGS, 2018) in the effort to expand the total water level predictions (Aretxabaleta et al., 2019).

## 2. Regional description

The Barnegat Bay-Little Egg Harbor (BBLEH) estuary is a back-barrier bay along the coast of New Jersey (Figure 1). It is a shallow (average depth around 1.5 m) bay connected to the ocean through three openings: Little Egg Inlet in the south, Barnegat Inlet in the center, and Point Pleasant Canal, a much smaller connection in the north of the bay. Offshore tidal amplitudes decrease slightly from 0.7 m in northern New York Bay to 0.6 m in central New Jersey. The southern sub-embayment (Little Egg Harbor) is more connected with the open ocean with tidal amplitudes ranging between 0.2 and 0.5 m, while the northern part (Barnegat Bay) has less exchange and tidal amplitudes are smaller than 0.2 m (Chant, 2001; Defne and Ganju, 2015).

## 3. Observational and model data

Water level observations from five stations in the BBLEH system (Table 1) and from two external coastal stations are used to determine transfer from ocean to bay. The bay stations started recording in October 2007, while Sandy Hook and Atlantic City are long-term NOAA water level stations operational since 1910 and 1911, respectively (Table 1). Wind observations were obtained from the NDBC buoy 44065 (New York Harbor Entrance) for the period 2008-2018.

We used numerical simulations of Barnegat Bay for the period March-September 2012 (Defne and Ganju, 2015) and October-December 2012 (Defne and Ganju, 2019) to obtain the spatial character of the water level response. The simulations used the Coupled Ocean Atmospheric Wave Sediment Transport modeling system (COAWST; Warner et al., 2010). The model resolution ranged from 40 m to 200 m, with the higher resolution located near complex geometry and around the inlets. The model is forced at the boundaries with tides from the ADCIRC tidal database for the western North Atlantic Ocean (Szpilka et al., 2016) and open-ocean forcing from subtidal water level and velocity from the ESPreSSO model (Wilkin and Hunter, 2013; http://www.myroms.org/espresso/) and COAWST US east coast forecast (Warner et al., 2010; https://catalog.data.gov/dataset/coawst-forecast-system-usgs-us-east-coast-and-gulf-of-mexico-experimentalf9168). Defne

and Ganju (2015) showed the numerical model solution had sufficient flow and elevation skill to characterize bay dynamics
under normal and storm conditions.

The impact in the bay of offshore forcing can be evaluated spectrally by estimating transfer functions in frequency space between observed water levels offshore (input) and in the bays (output). The transfer functions are calculated using a Hanning window with over-lapping (50%) data segments with a length of 29 days to provide estimates near the main tidal frequencies (Aretxabaleta et al., 2017). The uncertainty envelopes for the transfer function are estimated using the Bendat
and Piersol (1986) formulation.

## 4 Analytical Water Level Models

### 4.1 Offshore Impact on Bay Model

The impact of ocean sea level fluctuations in the bay can be explored with an analytical model of a generic bay system (Figure 2) consisting of multiple interconnected sub-embayments connected to the offshore by three separate inlets: Little
Egg Inlet, Barnegat Inlet, and Point Pleasant Canal. The model assumes that the bay water level responds as a level surface in each sub-embayment to ocean fluctuations, as local forcing in the bay is not included. The formulation is an extension of the approach proposed by Chuang and Swenson (1981) for a single inlet connecting to a bay and expanded by Wong and DiLorenzo (1988) to two connected bays and to multiple bays and inlets by Aretxabaleta et al. (2017). An analytical solution can be found for the entire system with expressions for all the connections in the system. The model solves the along-
channel depth-averaged momentum equation based on the balance between frictional effects and the elevation gradient between offshore and bay and the continuity equation for the bay/channel system based on the changing volume of the bays as water flows through the inlets. The model also allows the estimation of the effect of the breach in Mantoloking during Hurricane Sandy. An analytical solution can be found by dividing the entire system into 5 sub-embayments (based on constrictions inside the bay system) resulting in a system of equations that includes 13 equations and unknowns (Appendix
A).

$$\frac{\partial}{\partial t}\begin{pmatrix} u_1 \\ u_2 \\ u_3 \\ u_4 \\ u_5 \\ u_6 \\ u_7 \\ u_8 \end{pmatrix} = g \begin{pmatrix} 1/L_1 \\ 1/L_2 \\ 1/L_3 \\ 1/L_4 \\ 1/L_5 \\ 1/L_6 \\ 1/L_7 \\ 1/L_8 \end{pmatrix} \begin{pmatrix} \eta_o\phi_{LEI} - \eta_1 \\ \eta_1 - \eta_2 \\ \eta_2 - \eta_3 \\ \eta_o\phi_{BI} - \eta_3 \\ \eta_3 - \eta_4 \\ \eta_4 - \eta_5 \\ \eta_o\phi_{breach} - \eta_5 \\ \eta_o\phi_{PPC} - \eta_5 \end{pmatrix} - \begin{pmatrix} u_1 r_1/h_1 \\ u_2 r_2/h_2 \\ u_3 r_3/h_3 \\ u_4 r_4/h_4 \\ u_5 r_5/h_5 \\ u_6 r_6/h_6 \\ u_7 r_7/h_7 \\ u_8 r_8/h_8 \end{pmatrix} \qquad (1)$$

$$\begin{pmatrix} A_1 \\ A_2 \\ A_3 \\ A_4 \\ A_5 \end{pmatrix} \frac{\partial}{\partial t} \begin{pmatrix} \eta_1 \\ \eta_2 \\ \eta_3 \\ \eta_4 \\ \eta_5 \end{pmatrix} = \begin{pmatrix} h_1 W_1 u_1 - h_2 W_2 u_2 \\ h_2 W_2 u_2 - h_3 W_3 u_3 \\ h_3 W_3 u_3 + h_4 W_4 u_4 - h_5 W_5 u_5 \\ h_5 W_5 u_5 - h_6 W_6 u_6 \\ h_6 W_6 u_6 + h_7 W_7 u_7 + h_8 W_8 u_8 \end{pmatrix} \qquad (2)$$

where $g$ is the gravitational acceleration, $A_m$ is the surface area of sub-embayment $m$; $\eta_m$ the sea level in the $m$ sub-embayment; $\eta_o$ the sea level in the ocean; with $h_n$ the water depth; $W_n$ the width; $L_n$ the length, and $r_n$ the linear drag coefficient of channel $n$. $\phi_{LEI}$, $\phi_{BI}$, $\phi_{breach}$, $\phi_{PPC}$ are the linear frequency-dependent relationships between the water levels at offshore proxy stations (Sandy Hook or Atlantic City) and the water level just offshore of Little Egg Inlet, Barnegat Inlet, the breach at Mantoloking caused by Sandy, and Point Pleasant Canal.

Assuming $\eta = \tilde{\eta}e^{i\omega t}$ and $u = \tilde{u}e^{i\omega t}$, where $\tilde{\eta}$ and $\tilde{u}$, represent the magnitude of the water level and velocity oscillations, respectively. Then, to reduce the size of the equations, we can define $K_n = \frac{h_n W_n g}{L_n\left(\frac{r_n}{h_n}+i\omega\right)}$ for $n$=1 ,…, 8, as the relative contribution of each sub-embayment based on its geometric and frictional characteristics. Then, with the proper rearrangement, it yields:

$$\tilde{\eta_3} = \frac{K_4\eta_o\phi_{BI} + \dfrac{\dfrac{K_1 K_2 K_3 \tilde{\eta_o}\phi_{LEI}}{i\omega A_1 + K_1 + K_2}}{i\omega A_2 + K_2 + K_3 - \dfrac{K_2 K_2}{i\omega A_1 + K_1 + K_2}} + \dfrac{\dfrac{K_5 K_6 (K_7\phi_{breach} + K_8\phi_{PPC})\tilde{\eta_o}}{i\omega A_5 + K_6 + K_7 + K_8}}{i\omega A_4 + K_5 + K_6 - \dfrac{K_6 K_6}{i\omega A_5 + K_6 + K_7 + K_8}}}{i\omega A_3 + K_3 + K_4 + K_5 - \dfrac{K_3 K_3}{i\omega A_2 + K_2 + K_3 - \dfrac{K_2 K_2}{i\omega A_1 + K_1 + K_2}} - \dfrac{K_5 K_5}{i\omega A_4 + K_5 + K_6 - \dfrac{K_6 K_6}{i\omega A_5 + K_6 + K_7 + K_8}}}$$


The solution for the water level of the central sub-embayment can be used to recursively calculate the solutions for the rest of the sub-embayments:

$$\widetilde{\eta_2} = \frac{K_3\widetilde{\eta_3}+\frac{K_2K_1\widetilde{\eta_o}\phi_{LEI}}{i\omega A_1+K_1+K_2}}{i\omega A_2+K_2+K_3-\frac{K_2K_2}{i\omega A_1+K_1+K_2}} \qquad (4)$$

$$\widetilde{\eta_4} = \frac{K_5\widetilde{\eta_3}+\frac{K_6(K_7\phi_{breach}+K_8\phi_{PPC})\widetilde{\eta_o}}{i\omega A_5+K_6+K_7+K_8}}{i\omega A_4+K_5+K_6-\frac{K_6K_6}{i\omega A_5+K_6+K_7+K_8}} \qquad (5)$$


$$\widetilde{\eta_1} = \frac{K_1\widetilde{\eta_o}\phi_{LEI}+K_2\widetilde{\eta_2}}{i\omega A_1+K_1+K_2} \qquad (6)$$

$$\widetilde{\eta_5} = \frac{(K_7\phi_{breach}+K_8\phi_{PPC})\widetilde{\eta_o}+K_6\widetilde{\eta_4}}{i\omega A_5+K_6+K_7+K_8} \qquad (7)$$

The resulting expressions include all the sub-embayment and offshore exchanges under the same assumptions of the Chuang and Swenson (1981) model (e.g., no local influences, no overtopping).

**4.2 Local Wind Impact on Bay Model**

The contribution of local wind setup to the spatial distribution of water level inside the bay can be approximated following Wong and Moses-Hall (1998). The bay can be assumed to be a simple long well-mixed embayment for which the cross-bay gradients and vertical stratification can be ignored. The linearized vertically integrated momentum and mass conservation equations are:


$$\frac{\partial U}{\partial t} = -gh\frac{\partial \eta}{\partial x}+\frac{1}{\rho_0}(\tau_s-\tau_b) = -gh\frac{\partial \eta}{\partial x}+\tau_w-r\frac{U}{h} \qquad (8)$$

and

$$\frac{\partial U}{\partial x} = -\frac{\partial \eta}{\partial t} \qquad (9)$$

where $\eta$ is the water level in the bay; $U$ is the depth integrated along-bay velocity; $h$ is the water depth; $\tau_s$ and $\tau_b$ are the surface and bottom dynamic stresses, respectively; and $\rho_0$ is the water density. $\tau_w = {\tau_s}/{\rho_0}$ is the spatially invariant kinematic wind stress and $r$ is a linearized bottom friction.

Under the assumption of $\eta = \tilde{\eta}e^{i\omega t}$ and $u = \tilde{u}e^{i\omega t}$, where $\omega$ is the cyclic frequency, the resulting equation is:

$$\frac{\partial^2 \tilde{\eta}}{\partial x^2} + \tilde{\eta}\left(\frac{\omega^2}{gh} - \frac{i\omega r}{gh^2}\right) = \frac{\partial^2 \tilde{\eta}}{\partial x^2} + \tilde{\eta}k^2 = 0 \ . \qquad (10)$$

with boundary conditions $\eta(x = 0, \omega) = 0$ assuming no offshore forcing at the entrance (this assumption will be revisited in the next section) and $\frac{\partial \tilde{\eta}(x=L,\omega)}{\partial x} = \frac{\widetilde{\tau_w}(\omega)}{gh}$ assuming no flux at the head ($x=L$). $\widetilde{\tau_w}(\omega)$ represents the magnitude of the kinematic wind stress that results in water level fluctuations at a specific frequency.

The solution is:

$$\tilde{\eta}(\omega) = \frac{\widetilde{\tau_w}(\omega)\sin(kx)}{ghk\,\cos(kL)} \qquad (11)$$

$$\text{where } k = \left(\frac{\omega^2}{gh} - \frac{i\omega r}{gh^2}\right)^{1/2}. \qquad (12)$$

The wavenumber $k$ determines the spatial response of the transfer between the wind stress and the bay water level. The imaginary wavenumber part leads to exponential decay based on the frictional characteristics. The magnitude of water level is obtained from Equation 11, while the ratio of real to imaginary parts provides information about the phase lag between wind stress and water level.

### 4.3 Combining Local and Remote Effects

The local and remote effects can be combined in following the approach by Wong and Moses-Hall (1998). While bays can exhibit complex spatial responses to wind forcing especially in terms of currents (Csanady, 1973; Hunter and Hearn, 1987; Cioffi et al., 2005), the basic response can be summarized as the sum of local (wind) and remote (surge) forcings. The boundary condition for the local wind effect can be altered to account for the influence of offshore water level, $\eta_o$. The resulting model is a modification of the wind effect model that considers the analytical offshore influence in Section 4.1. In a

system with a single inlet, the solution can be simply stated as in Wong and Moses-Hall (1998):

$$\tilde{\eta}(\omega) = \frac{\tilde{\tau_w}(\omega)\,sin(kx)}{ghk\,cos(kL)} + \widetilde{\eta_o}(\omega)\frac{cos(k(L-x))}{cos(kL)} \qquad (13)$$

In a system with multiple connections with the offshore, the solution can be more complex. One limitation of the approach is that it utilizes a linear friction approximation. To produce a better approximation that takes into account the complex

frictional conditions of the bay (e.g., varying geometry, diverse bottom conditions, enhanced attenuation over submerged vegetation), we can take a numerical solution of the bay that resolves the tidal and sub-tidal water level conditions under realistic friction and adjust the spatial distribution of the transfer from offshore accordingly. As most of the water level variability in the bay is associated with the $M_2$ semidiurnal tidal constituent (Figure 3) and the distribution of the tide has been properly validated in the numerical simulations (Defne and Ganju, 2015), we can take the spatial distribution of the $M_2$

tidal amplitude as a proxy for the internal frictional effects in the bay. Bottom friction caused by both wind driven and tidal effects is considered in the numerical simulations. By adjusting the water level based on the numerical $M_2$ spatial distribution, we are approximating the complete frictional characteristics of the bay. The adjustment is applied to each of the sub-embayments following the expression:

$$\frac{\hat{\eta}_j}{\eta_o}(x,\omega) = 1 - \frac{\left(1-\frac{\eta_j}{\eta_o}(\omega)\right)\left(1-\frac{\eta\left(x,\omega_{M_2}\right)}{\eta\left(offshore,\omega_{M_2}\right)}\right)}{\left(1-\frac{\eta_j}{\eta_o}(\omega_{M_2})\right)} \qquad (14)$$

where $\eta_j/\eta_o\,(\omega)$ is the transfer coefficient of the sub-embayment $i$ (single value) at frequency $\omega$, $\eta(x,\omega_{M_2})$ is the amplitude of the $M_2$ tidal fluctuations from the numerical model solution (spatially variable), and $\hat{\eta}_j/\eta_o\,(x,\omega)$ is the spatially variable adjusted transfer coefficient for sub-embayment $j$. The resulting adjusted transfer coefficients provide estimates of the spatial changes not only between adjacent sub-embayments but also inside each of the sub-embayments. The

local wind effects on bay water level can be added to the impact from offshore fluctuations to obtain a combined local and remote water level response estimate. In cases with simultaneous presence of wind and offshore level fluctuations, the system can respond in a weakly nonlinear manner and departures from the presented basic addition of the process are expected.

**5 Results**

**5.1 Offshore transfer to bay**

The maximum energy in water level spectra (Figure 3, Table 2) was associated with the $M_2$ semidiurnal tidal constituent for the offshore proxies (SH and AC) and at the stations TUC and ETH in the southern part of the BBLEH area. For the locations in Barnegat Bay (WAR, SEH, MAN), maximum energy was in the low frequency band. Large spectral energy also occurred in the other semidiurnal tidal frequencies ($S_2$ and $N_2$), the diurnal frequencies ($O_1$ and $K_1$), the storm band (periods

between 2 and 5 days), and the low frequency band (Table 2). The energy in the remaining bands exhibited average fluctuations less than 0.03 m in size offshore, while in the bay fluctuations were less than 0.01 m.

Transfer functions between Atlantic City (AC) and the five stations inside the bay (Figure 4) for the longest available length of record showed a north to south gradient. The transfer of the offshore fluctuations was 50-80% at periods between 2 and 5 days (storm band) except at Tuckerton (TUC; over 95%). The transfers at diurnal periods were about 35% for the three

Barnegat Bay stations (WAR, SEH, MAN), about 45% in Little Egg Harbor (ETH), and 80% in Great Bay (TUC). For frequencies associated with the semidiurnal tides, the transfers were even more attenuated with values about 15% (between 14 and 16%) inside Barnegat, 30-35% at ETH, and 60-70% at TUC. As the numerical model solution was only available for the period March-December 2012, the long-term (2007-2018) transfers were compared with shorter-term observations. The transfers were similar (within the uncertainty envelopes for each station, not shown) for both datasets at most frequencies

except at Mantoloking (MAN) that showed enhanced transfers for periods between one and five days in the 2012 record and at Seaside Heights (SEH), where transfers in the storm band were slightly attenuated during 2012. Transfer estimates using Sandy Hook (SH) as the offshore proxy instead of Atlantic City produced similar results (not shown). The transfer between stations AC and SH on the open coast (proxies for offshore fluctuations) has been shown to be close to one (Wong and Wilson, 1984; Aretxabaleta et al., 2014), confirming that the offshore forcing at all three inlets is about the same.

Transfers estimated from the numerical model solution (Figure 5) showed similar magnitudes to the observed transfers (within uncertainty envelopes provided by the Bendat and Piersol (1986) formulation) at most frequencies. The observed and modeled transfer at diurnal and semi-diurnal transfers were similar (within a few percentage points) at all stations except the model overestimated the semidiurnal transfer at TUC. Differences between model and observed estimates at MAN and SEH only were significant at frequencies that contained minimal energy. The model reproduced the enhanced transfer in the storm

band at Mantoloking during 2012, suggesting a physical mechanism for the change that the model was able to capture but remains unexplained. The likely explanation is that the location of the Azores-Bermuda high-pressure system over the Atlantic in 2012 (Mattingly et al., 2012), associated with the negative phase of the North Atlantic Oscillation, resulted in average winds that lined up with the axis of the Bay and caused enhanced wind setup in the northern part of the bay. The

model overestimated the transfer at ETH in the storm band and underestimated the low-frequency transfer at Waretown. The likely cause for some of the discrepancies, especially at low frequencies, is the relatively short length of the available record.

The analytical model of offshore impact that considered five sub-embayments (Section 4.1) was fit to the observed transfers to obtain an estimate of linear friction. The fit considered the unevenly distributed energy spectra (Figure 3) with adjusted weight to the semidiurnal and low-frequency components. The resulting friction was r = 0.021 m/s. The associated frictional adjustment time, $t_{fr} = {h}/{r}$, was about 1-5 minutes depending on the depth of the inlet. The analytical curves (Figure 6) matched the observed transfer function shape at most frequencies. The analytical model with five sub-embayment domains captured the north-south spatial differences in transfer. The analytical model for the central Barnegat Bay sub-embayment ($A_3$) approximated the transfer estimates from observations at Waretown at most frequencies (less than 5 % in the storm band). The analytical model for Great Bay ($A_1$) adequately matched observed transfers at Tuckerton at diurnal and semidiurnal frequencies (less than 5 % difference) but underestimated the transfer in the storm band (model estimates about 90%, while observations were above 95%). Meanwhile, the analytical model for Little Egg Harbor (A2) matched the observed transfers at ETH within the uncertainty envelope, except for a slight under-prediction at diurnal frequencies (less than 5%). The observed transfers at the northern stations (MAN and SEH) were reproduced by the analytical model ($A_4$, $A_5$ respectively) at diurnal and semidiurnal tides, but were under-predicted for the higher storm band frequencies (5-10% less transfer in frequencies close to 2-day periods) and over-predicted at low frequencies (about 10% differences). The analytical model was used to explore the effect on transfer of the breach ($U_7$) at Mantoloking that opened during Hurricane Sandy. The transfers were so minimally affected that the curves are indistinguishable with only a negligible enhancement (<0.2%) in transfer in the northern most sub-embayment ($A_5$). The breach was too small and shallow for any significant volume transport to occur that would affect the large bay.

## 5.2 Local Wind Influence

The spectrum of the along-bay component (rotated 20 degrees) of the wind (Figure 7a) from the offshore buoy NDBC 44065 (ten years, 2008-2018) showed high energies in the storm band (2-5 days) and in low frequencies. The largest single peak of energy was associated with 24-hour period oscillations likely associated with sea breeze and matched energy values at 5-day frequencies. There was a small peak at inertial frequencies.

The local wind contribution to water level setup inside the bay was approximated using the Wong and Moses-Hall (1998) approach (Section 4.2). The resulting formulation showed largest setup magnitudes near the head of the bay (e.g., northern part with wind blowing from the south) with a decay as distance from head increased (Figure 7b,c). The magnitude of the setup depended on the magnitude of the linear friction with less setup under stronger friction (Figure 7b,c). The setup responded exponentially to fetch (distance) except under long duration and low friction conditions, which was predominantly

linear (Figure 7b). The frictional control was less important at higher frequencies (Figure 7c). As frequency increased there was less wind energy (Figure 7a), so the frictional control is mostly important for low frequency and storm band wind fluctuations.

The resulting effect of the wind setup (or set-down) was small (less than 0.1 m with an along-bay wind stress of 0.1 Pa) for most of the domain (Figure 8). The estimate assumed a linear friction of the same magnitude as in Section 5.1 ($r$=0.021 m/s). Under persistent wind stress of 0.1 Pa (about 8 m/s wind speed) in the along-bay direction, the resulting setups varied depending on the frequency considered. Setup magnitudes over 0.2 m were estimated for the 5-day period wind (Figure 8c), while under half of that magnitude was achieved for the 2-day persistent wind (Figure 8b), and much smaller water level setup (peak smaller than 0.1 m) was estimated for the sea breeze (Figure 8a). During extreme events like Hurricane Sandy, under intense wind stress, two additional effects should be considered: the depth of the bay increases by the transfer of offshore surge resulting in altered setup response (Section 4.2), and the frictional effect is enhanced (a larger linear friction would be needed) by the presence of wave-induced roughness.

## 6 Discussion

### 6.1 Spatially Variable Water Level Transfer

Following the approach described in Section 4.3, estimates of spatially variable water level impact from offshore can be calculated (Figure 9). The $M_2$ tidal constituent transfer (Figure 9a) showed a large north to south gradient with values going from around 10% in the north to over 80% in the vicinity of Little Egg Inlet. The role of Barnegat Inlet in enhancing tidal transfer was greatly reduced as most of the tide was attenuated in the inlet. The contribution of Point Pleasant Canal was also small as expected from the tidal amplitudes (Chant, 2001; Defne and Ganju, 2015). The transfer in the storm band of 2-day fluctuations (Figure 9b) also showed a strong north-south gradient with values about 50% in Barnegat Bay, around 70-80% in Little Egg Harbor and larger values in Great Bay. The 5-day offshore fluctuations were transferred more efficiently into the bay (Figure 9c) with values over 70% in the entire bay, reaching 80-90% in Little Egg Harbor, and over 90% in Great Bay. Both storm band transfer estimates were controlled by the exchange through Little Egg Inlet, with very local transfer enhancements in the vicinity of Barnegat Inlet and Point Pleasant Canal. While the presented estimates used Atlantic City as offshore proxy, similar results were obtained when Sandy Hook was used as the offshore reference (as expected from Aretxabaleta et al., 2014).

When the magnitude of the fluctuations associated with a specific storm are available $\eta_o$, then an estimate of the average water level in the bay during the storm can be obtained. For instance, for Hurricane Sandy the offshore surge associated with the storm was of the order of 2-3 m. Considering that the storm lasted for over a day, the water level transfer would have

been above 50% in Barnegat Bay and above 70% in Little Egg Harbor. The resulting surge estimate in the bay was between 1 and 2 m just considering the exchange through the existing inlets. There was reported overtopping of the barrier island during the storm (McKenna et al., 2016) that might have further increased water level in the bay that the proposed method does not consider.

The wind setup effect inside the bay due to local wind can also be estimated for Hurricane Sandy using the approach in Section 4.2. Maximum wind stress during the storm was about 1 Pa. To obtain a maximum effect (worst-case scenario) the wind was assumed to be persistently in the along-bay direction and that maximum stress was maintained for the duration of the storm. The maximum resulting water level considering the Wong and Moses-Hall method is linear with regard to wind stress magnitude (Figure 7b) and would have been 10 times larger than the setup in Figure 8b. The maximum wind setup would have been between 1 and 2 m, which was of the same order of magnitude as the surge produced from offshore sources. The cross-bay contribution to the wind setup during Sandy was comparatively small as wind direction was predominantly along-bay. Surge estimates from simple analytical formulations (State Committee for the Zuiderzee, 1926; Pugh, 1987) that do not consider storm duration produce similar magnitude results and are also dependent on the frictional response of the bay. Nonlinear interactions between local and remote effects may alter the total bay response but these effects are likely second order.

### 6.2 Transfer Based on Tidal Database

The approach thus far was based on the combination of observations, analytical models, and numerical models. In many systems, long-term observations that allow for the estimation of transfer coefficients might not be available. Also, numerical solutions of back-barrier bay systems tend to be computationally expensive and might not be available for the period of interest. We propose a relatively simpler approach for some of these systems based on the availability of high-resolution tidal solutions for the system. The EC2015 ADCIRC tidal database (Szpilka et al., 2016) showed sufficient resolution (down to 13m in some areas) in many bays along the east coast of the United States to resolve the tidal conditions with skill when compared to NOAA CO-OPS stations and historic International Hydrographic Organization (IHO) data. The EC2015 tidal database provides estimates for 37 tidal constituents. Based on those constituents and assuming that the totality of the offshore fluctuations at zero frequency reach the interior of the bay, an estimate can be provided for the storm band frequencies. A weighted least squares interpolation in the frequency domain was performed based on the $M_4$, $K_2$, $S_2$, $L_2$, $M_2$, $N_2$, $K_1$, $P_1$, $O_1$, $Q_1$ tidal amplitudes ratios between each point of the ADCIRC domain inside the bay and a point in the offshore. Higher weight was given to zero frequency to nudge toward 100% transfer at zero frequency. Estimates were calculated based on multiple locations inside the bay and average to achieve a more robust calculation and also obtain an approximation to the uncertainty associated with the estimate.

The resulting transfer estimates (Figure 10) exhibited the same general spatial patterns shown in the previous estimates (Figure 9) with slight differences. Some of the smaller features present in the COAWST numerical solution (Defne and Ganju, 2015) were not present in the ADCIRC EC2015 domain. The $M_2$ transfer estimate based on the tidal database (Figure 10a) presented approximately the same magnitudes in most areas (average difference less than 3%). The 5-day transfer (Figure 10c) was also comparable to the solution described in Section 6.1 with values over 70% in the entire domain and the southern areas exceeding 90% transfer. The 2-day transfer from ADCIRC (Figure 10b) was 5-10% higher than the direct estimates (Figure 9b). One of the benefits of the ADCIRC approach was the possibility of providing an approximation to the uncertainty (Figure 11). The uncertainty estimate of the $M_2$ transfer was about 1-2% (Figure 11a) with higher values in the southern part of the domain. The 2-day transfer uncertainty (Figure 11b) was above 4% in Barnegat Bay in areas of larger discrepancy between the ADCIRC and complete approaches. The uncertainty estimates in the 5-day offshore water level transfer (Figure 11c) in the northern part of the domain did not exceed 2.5%.

The magnitude of the difference between the ADCIRC tidal database approach and the complete method highlighted in Section 4.3 was of the same order of magnitude or even smaller than the difference between observations and analytical model (Figure 6) or between observed and numerical modeled transfers (Figure 5). This result emphasizes the validity of using the tidal database to calculate offshore transfer estimates, especially when water level observations inside the bay or numerical solutions are not available.

The effect of local wind setup will also need to be added to the ADCIRC-based estimate, especially during severe storms. The approach discussed in Section 5.2 or even a simpler surge calculation (e.g., from the steady state vertically averaged momentum equations, as in Pugh (1987), from the traditional report of the State Committee for the Zuiderzee (1926), or the updated frequency domain equivalent from Reef et al., 2018) could be used and the resulting elevation could be added to the offshore transfer estimate obtain based on the ADCIRC tides. Thus, the production of bay water level predictions will require accurate wind forecast products and the quantification of the nonlinear interaction between local and remote effects.

**6.3 Validity for Flooding Hazard Assessments**

The method presented offers a new methodology for coastal hazards assessment and risk analysis. While many methodologies are being used for open-coast regions (Thieler and Hammar-Klose, 1999; Kunreuther et al., 2000), vulnerability evaluation to coastal hazards in back-barrier bays remains under-developed. Evaluating bay hazards usually requires expensive computational simulations at appropriate high resolutions to characterize the spatial and temporal effects. The method presented here, using existing ADCIRC results, provides a less expensive approach that is able to properly estimate the spatial differences in vulnerability in response to flooding at different time scales (e.g., perigean spring tides, storms of different duration). It provides guidance for planning in response to "nuisance" flooding at a relatively low cost. It

can be expanded to all back-barriers without the need to simulate each storm in each embayment, while applying a consistent methodology.

Careful consideration needs to be given to the estimation of coastal hazards especially for the forecast of intense storm effects. The inclusion of meticulously validated methodologies that consider both offshore influences (e.g., using the transfer estimated from ADCIRC tides) and local wind setup (e.g., Wong and Moses-Hall, 1998; Reef et al., 2018) is necessary. Skill assessment of storm hazard estimates using adequate observations is critical to avoid producing under- or over-predictions of flooding and inundation.

As part of the general needs for hazard assessment (Ludwig et al., 2018), the important hazard characteristics that decision makers require include spatial extent, duration, and magnitude. The proposed methodology provides an approximation to both the area extent and magnitude, and also variations based on storm duration. Additionally, the fact that uncertainty estimates accompany the vulnerability provided by the present method enhances the potential value to decision makers. The extension to other bays in the United States will be included as part of the U. S. Geological Survey Coastal Change Hazards
portal (USGS, 2018).

**7 Summary**

The results presented here demonstrate a strategy for estimating the impact of offshore sea level and local wind setup in back-barrier bay water levels. The transfer estimates of offshore to bay water level used a combination of observations, analytical models based on appropriate simplifications of the bay system, and numerical simulations that provide the needed
spatial distribution and more realistic frictional control.

The resulting maps of water level response to offshore forcing showed larger attenuation of the relatively higher frequency fluctuations such as the semidiurnal tides. Smaller transfers were associated with shorter duration storms than longer duration storms and transfer was most spatially uniform for storms of long duration. The description of the magnitude and spatial dependence of transfer on storm duration will assist planning for flooding in back-barrier bays.

In the specific case of the Barnegat Bay-Little Egg Harbor system, larger transfers were estimated for the southern embayments (Great Bay and Little Egg Harbor) when compared to Barnegat Bay. The reason for the difference was the dominant role of Little Egg Inlet (wider and deeper) in controlling the exchange between the offshore and bay systems. During relatively small storms, the contribution of local wind to bay water level setup was smaller than the transfer from offshore fluctuations. During intense events, like hurricanes, local wind setup was of the same order of magnitude or even

larger than offshore influences depending on wind magnitude and especially the relative angle of the wind with respect to bay orientation.

We introduced two approaches depending on the availability of observations and numerical solutions. The less data-requiring approach based on the ADCIRC tidal database provides spatial offshore transfer estimates and measures of uncertainty. In both cases, the inclusion of the local wind setup could be achieved based on simple surge analytical estimates.

The approach that includes an analytical model allows for a simple tool to study the response of back-barrier bay systems to alternative conditions and forcing (e.g., geomorphic changes, changing duration of storms, sea level rise).

The proposed method represents an effective and inexpensive approach to flooding hazard evaluation in back-barrier bays and inland waters. The method provides detailed spatial estimates of vulnerabilities and uncertainties that could be an intuitive tool for coastal managers.

**Appendix A**

The offshore impact in the bays water level can be approximated with an analytical model that solves the linearized depth-averaged momentum equations. The system of equations for an idealized simplification of Barnegat Bay (Figure 2) that includes 5 sub-embayments (based on constrictions inside the bay system) consists of 13 equations and unknowns.


$$
\frac{\partial}{\partial t}
\begin{pmatrix} u_1 \\ u_2 \\ u_3 \\ u_4 \\ u_5 \\ u_6 \\ u_7 \\ u_8 \end{pmatrix}
= g
\begin{pmatrix} 1/L_1 \\ 1/L_2 \\ 1/L_3 \\ 1/L_4 \\ 1/L_5 \\ 1/L_6 \\ 1/L_7 \\ 1/L_8 \end{pmatrix}
\begin{pmatrix} \eta_o \phi_{LEI} - \eta_1 \\ \eta_1 - \eta_2 \\ \eta_2 - \eta_3 \\ \eta_o \phi_{BI} - \eta_3 \\ \eta_3 - \eta_4 \\ \eta_4 - \eta_5 \\ \eta_o \phi_{breach} - \eta_5 \\ \eta_o \phi_{PPC} - \eta_5 \end{pmatrix}
-
\begin{pmatrix} u_1 r_1/h_1 \\ u_2 r_2/h_2 \\ u_3 r_3/h_3 \\ u_4 r_4/h_4 \\ u_5 r_5/h_5 \\ u_6 r_6/h_6 \\ u_7 r_7/h_7 \\ u_8 r_8/h_8 \end{pmatrix}
\qquad (\text{Ap1})
$$

$$
\begin{pmatrix} A_1 \\ A_2 \\ A_3 \\ A_4 \\ A_5 \end{pmatrix} \frac{\partial}{\partial t} \begin{pmatrix} \eta_1 \\ \eta_2 \\ \eta_3 \\ \eta_4 \\ \eta_5 \end{pmatrix} = \begin{pmatrix} h_1 W_1 u_1 - h_2 W_2 u_2 \\ h_2 W_2 u_2 - h_3 W_3 u_3 \\ h_3 W_3 u_3 + h_4 W_4 u_4 - h_5 W_5 u_5 \\ h_5 W_5 u_5 - h_6 W_6 u_6 \\ h_6 W_6 u_6 + h_7 W_7 u_7 + h_8 W_8 u_8 \end{pmatrix} \tag{Ap2}
$$

$\phi_{LEI}, \phi_{BI}, \phi_{breach}, \phi_{PPC}$ are linear frequency-dependent relationships between the water levels at offshore proxy stations (Sandy Hook or Atlantic City) and the water level just offshore of Little Egg Inlet, Barnegat Inlet, the breach at Mantoloking

caused by Sandy, and Point Pleasant Canal.

By performing Fourier transforms on the momentum equations ($\eta = \tilde{\eta} e^{i\omega t}$ and $u = \tilde{u} e^{i\omega t}$, where $\tilde{\eta}$ and $\tilde{u}$, represent the magnitude of the water level and velocity oscillations, respectively), we obtain:

$$
i\omega \begin{pmatrix} \tilde{u}_1 \\ \tilde{u}_2 \\ \tilde{u}_3 \\ \tilde{u}_4 \\ \tilde{u}_5 \\ \tilde{u}_6 \\ \tilde{u}_7 \\ \tilde{u}_8 \end{pmatrix} = g \begin{pmatrix} 1/L_1 \\ 1/L_2 \\ 1/L_3 \\ 1/L_4 \\ 1/L_5 \\ 1/L_6 \\ 1/L_7 \\ 1/L_8 \end{pmatrix} \begin{pmatrix} \tilde{\eta}_o \phi_{LEI} - \tilde{\eta}_1 \\ \tilde{\eta}_1 - \tilde{\eta}_2 \\ \tilde{\eta}_2 - \tilde{\eta}_3 \\ \tilde{\eta}_o \phi_{BI} - \tilde{\eta}_3 \\ \tilde{\eta}_3 - \tilde{\eta}_4 \\ \tilde{\eta}_4 - \tilde{\eta}_5 \\ \tilde{\eta}_o \phi_{breach} - \tilde{\eta}_5 \\ \tilde{\eta}_o \phi_{PPC} - \tilde{\eta}_5 \end{pmatrix} - \begin{pmatrix} \tilde{u}_1 r_1 / h_1 \\ \tilde{u}_2 r_2 / h_2 \\ \tilde{u}_3 r_3 / h_3 \\ \tilde{u}_4 r_4 / h_4 \\ \tilde{u}_5 r_5 / h_5 \\ \tilde{u}_6 r_6 / h_6 \\ \tilde{u}_7 r_7 / h_7 \\ \tilde{u}_8 r_8 / h_8 \end{pmatrix} \tag{Ap3}
$$

and then:

$$\begin{pmatrix} \widetilde{u_1} \\ \widetilde{u_2} \\ u_3 \\ u_4 \\ u_5 \\ u_6 \\ \widetilde{u_7} \\ u_8 \end{pmatrix} = g \begin{pmatrix} \dfrac{\widetilde{\eta_o}\phi_{LEI} - \widetilde{\eta_1}}{\widetilde{\eta_1} - \widetilde{\eta_2}} \\ \widetilde{\eta_2} - \eta_{3} \\ \dfrac{\widetilde{\eta_o}\phi_{BI} - \widetilde{\eta_3}}{\widetilde{\eta_3} - \widetilde{\eta_4}} \\ \eta_4 - \eta_5 \\ \dfrac{\widetilde{\eta_o}\phi_{breach} - \widetilde{\eta_5}}{\eta_o \phi_{PPC} - \eta_5} \end{pmatrix} \Bigg/ \left[ \begin{pmatrix} L_1 \\ L_2 \\ L_3 \\ L_4 \\ L_5 \\ L_6 \\ L_7 \\ L_8 \end{pmatrix} \begin{pmatrix} i\omega + {}^{r_1}\!/\!_{h_1} \\ i\omega + {}^{r_2}\!/\!_{h_2} \\ i\omega + {}^{r_3}\!/\!_{h_3} \\ i\omega + {}^{r_4}\!/\!_{h_4} \\ i\omega + {}^{r_5}\!/\!_{h_5} \\ i\omega + {}^{r_6}\!/\!_{h_6} \\ i\omega + {}^{r_7}\!/\!_{h_7} \\ i\omega + {}^{r_8}\!/\!_{h_8} \end{pmatrix} \right] \qquad (Ap4)$$

Performing the Fourier transform on the continuity equations (Eq. Ap2) and substituting the velocity values from

Eq. Ap4, we obtain:

$$i\omega \begin{pmatrix} A_1 \, \widetilde{\eta_1} \\ A_2 \, \eta_2 \\ A_3 \, \eta_3 \\ A_4 \, \eta_4 \\ A_5 \, \eta_5 \end{pmatrix} =$$

$$g \begin{pmatrix} \dfrac{h_1 W_1 \left(\widetilde{\eta_o}\phi_{LEI} - \widetilde{\eta_1}\right)}{L_1\left(i\omega + {}^{r_1}\!/\!_{h_1}\right)} - \dfrac{h_2 W_2 \left(\widetilde{\eta_1} - \widetilde{\eta_2}\right)}{L_2\left(i\omega + {}^{r_2}\!/\!_{h_2}\right)} \\[3mm] \dfrac{h_2 W_2 \left(\widetilde{\eta_1} - \widetilde{\eta_2}\right)}{L_2\left(i\omega + {}^{r_2}\!/\!_{h_2}\right)} - \dfrac{h_3 W_3 \left(\widetilde{\eta_2} - \widetilde{\eta_3}\right)}{L_3\left(i\omega + {}^{r_3}\!/\!_{h_3}\right)} \\[3mm] \dfrac{h_3 W_3 \left(\widetilde{\eta_2} - \widetilde{\eta_3}\right)}{L_3\left(i\omega + {}^{r_3}\!/\!_{h_3}\right)} + \dfrac{h_4 W_4 \left(\widetilde{\eta_o}\phi_{BI} - \widetilde{\eta_3}\right)}{L_4\left(i\omega + {}^{r_4}\!/\!_{h_4}\right)} - \dfrac{h_5 W_5 \left(\widetilde{\eta_3} - \widetilde{\eta_4}\right)}{L_5\left(i\omega + {}^{r_5}\!/\!_{h_5}\right)} \\[3mm] \dfrac{h_5 W_5 \left(\widetilde{\eta_3} - \widetilde{\eta_4}\right)}{L_5\left(i\omega + {}^{r_5}\!/\!_{h_5}\right)} - \dfrac{h_6 W_6 \left(\widetilde{\eta_4} - \widetilde{\eta_5}\right)}{L_6\left(i\omega + {}^{r_6}\!/\!_{h_6}\right)} \\[3mm] \dfrac{h_6 W_6 \left(\widetilde{\eta_4} - \widetilde{\eta_5}\right)}{L_6\left(i\omega + {}^{r_6}\!/\!_{h_6}\right)} + \dfrac{h_7 W_7 \left(\widetilde{\eta_o}\phi_{breach} - \widetilde{\eta_5}\right)}{L_7\left(i\omega + {}^{r_7}\!/\!_{h_7}\right)} + \dfrac{h_8 W_8 \left(\widetilde{\eta_0}\phi_{PPC} - \widetilde{\eta_5}\right)}{L_8\left(i\omega + {}^{r_8}\!/\!_{h_8}\right)} \end{pmatrix}$$

Then, we can define $K_n = \dfrac{h_n W_n g}{L_n\left(\frac{r_n}{h_n} + i\omega\right)}$ for each channel $n=1,\dots,8$ and with the proper rearrangement:

$$\begin{pmatrix} i\omega A_1 \widetilde{\eta_1} = K_1 \widetilde{\eta_o} \phi_{LEI} - K_1 \widetilde{\eta_1} - K_2 \widetilde{\eta_1} + K_2 \widetilde{\eta_2} \\ i\omega A_2 \widetilde{\eta_2} = K_2 \widetilde{\eta_1} - K_2 \widetilde{\eta_2} - K_3 \widetilde{\eta_2} + K_3 \widetilde{\eta_3} \\ i\omega A_3 \widetilde{\eta_3} = K_3 \widetilde{\eta_2} - K_3 \widetilde{\eta_3} + K_4 \widetilde{\eta_o} \phi_{BI} - K_4 \widetilde{\eta_3} - K_5 \widetilde{\eta_3} + K_5 \widetilde{\eta_4} \\ i\omega A_4 \widetilde{\eta_4} = K_5 \widetilde{\eta_3} - K_5 \widetilde{\eta_4} - K_6 \widetilde{\eta_4} + K_6 \widetilde{\eta_5} \\ i\omega A_5 \widetilde{\eta_5} = K_6 \widetilde{\eta_4} - K_6 \widetilde{\eta_5} + K_7 \widetilde{\eta_o} \phi_{breach} - K_7 \widetilde{\eta_5} + K_8 \widetilde{\eta_o} \phi_{PPC} - K_8 \widetilde{\eta_5} \end{pmatrix} \qquad \text{(Ap6)}$$

The system of equations can be solved by substitution.

$$\widetilde{\eta_5} = \frac{(K_7 \phi_{breach} + K_8 \phi_{PPC})\widetilde{\eta_o} + K_6 \widetilde{\eta_4}}{i\omega A_5 + K_6 + K_7 + K_8} \qquad \text{(Ap7)}$$

$$\widetilde{\eta_4} = \frac{K_5 \widetilde{\eta_3} + \frac{K_6(K_7 \phi_{breach} + K_8 \phi_{PPC})\widetilde{\eta_o}}{i\omega A_5 + K_6 + K_7 + K_8}}{i\omega A_4 + K_5 + K_6 - \frac{K_6 K_6}{i\omega A_5 + K_6 + K_7 + K_8}} \qquad \text{(Ap8)}$$

$$\widetilde{\eta_1} = \frac{K_1 \widetilde{\eta_o} \phi_{LEI} + K_2 \widetilde{\eta_2}}{i\omega A_1 + K_1 + K_2} \qquad \text{(Ap9)}$$

$$\widetilde{\eta_2} = \frac{K_3 \widetilde{\eta_3} + \frac{K_2 K_1 \widetilde{\eta_o} \phi_{LEI}}{i\omega A_1 + K_1 + K_2}}{i\omega A_2 + K_2 + K_3 - \frac{K_2 K_2}{i\omega A_1 + K_1 + K_2}} \qquad \text{(Ap10)}$$

and finally,

$$\widetilde{\eta_3} = \frac{K_4 \eta_o \phi_{BI} + \frac{\frac{K_1 K_2 K_3 \widetilde{\eta_o} \phi_{LEI}}{i\omega A_1 + K_1 + K_2}}{i\omega A_2 + K_2 + K_3 - \frac{K_2 K_2}{i\omega A_1 + K_1 + K_2}} + \frac{\frac{K_5 K_6(K_7 \phi_{breach} + K_8 \phi_{PPC})\widetilde{\eta_o}}{i\omega A_5 + K_6 + K_7 + K_8}}{i\omega A_4 + K_5 + K_6 - \frac{K_6 K_6}{i\omega A_5 + K_6 + K_7 + K_8}}}{i\omega A_3 + K_3 + K_4 + K_5 - \frac{K_3 K_3}{i\omega A_2 + K_2 + K_3 - \frac{K_2 K_2}{i\omega A_1 + K_1 + K_2}} - \frac{K_5 K_5}{i\omega A_4 + K_5 + K_6 - \frac{K_6 K_6}{i\omega A_5 + K_6 + K_7 + K_8}}}$$
$$\text{(Ap11)}$$

As it only depends on the offshore water level, the solution for the water level of the central sub-embayment (Ap11) can be used to recursively calculate the solutions for the rest of the sub-embayments by substituting into Equations Ap7, AP8, Ap9, and Ap10.

**Author Contribution**

A.L. Aretxabaleta developed the methodology and wrote most of the manuscript. N.K. Ganju and R.P. Signell suggested improvements to the methodology. Z. Defne contributed to the figure generation, the COAWST simulation analysis, and the discussion of results. All authors contributed to the final version.

**Competing Interests**

The authors declare that they have no conflict of interest.

**Disclaimer**

Use of firm and product names is for descriptive purposes only and does not imply endorsement by the U.S. Government.

**Acknowledgments**

The authors thank Brad Butman for helpful comments on the manuscript. This work was supported by the U.S. Geological Survey, Coastal and Marine Hazards/Resources Program. The hydrodynamic model outputs used in this study are available from Defne and Ganju (2019). The numerical model is the open source model COAWST (Warner et al., 2019) available

from   https://code.usgs.gov/coawstmodel/COAWST after registration with J. C. Warner (jcwarner@usgs.gov) at the U.S. Geological   Survey.   The   data   used   are   listed   in   the   references,   tables,   and   the   repository   at http://waterdata.usgs.gov/nwis/inventory/?site_no=xxx, where xxx stands for the USGS station number in Table 1. The paper benefited from conversations with Chris Sherwood, Tarandeep Kalra, and John Warner from USGS.

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

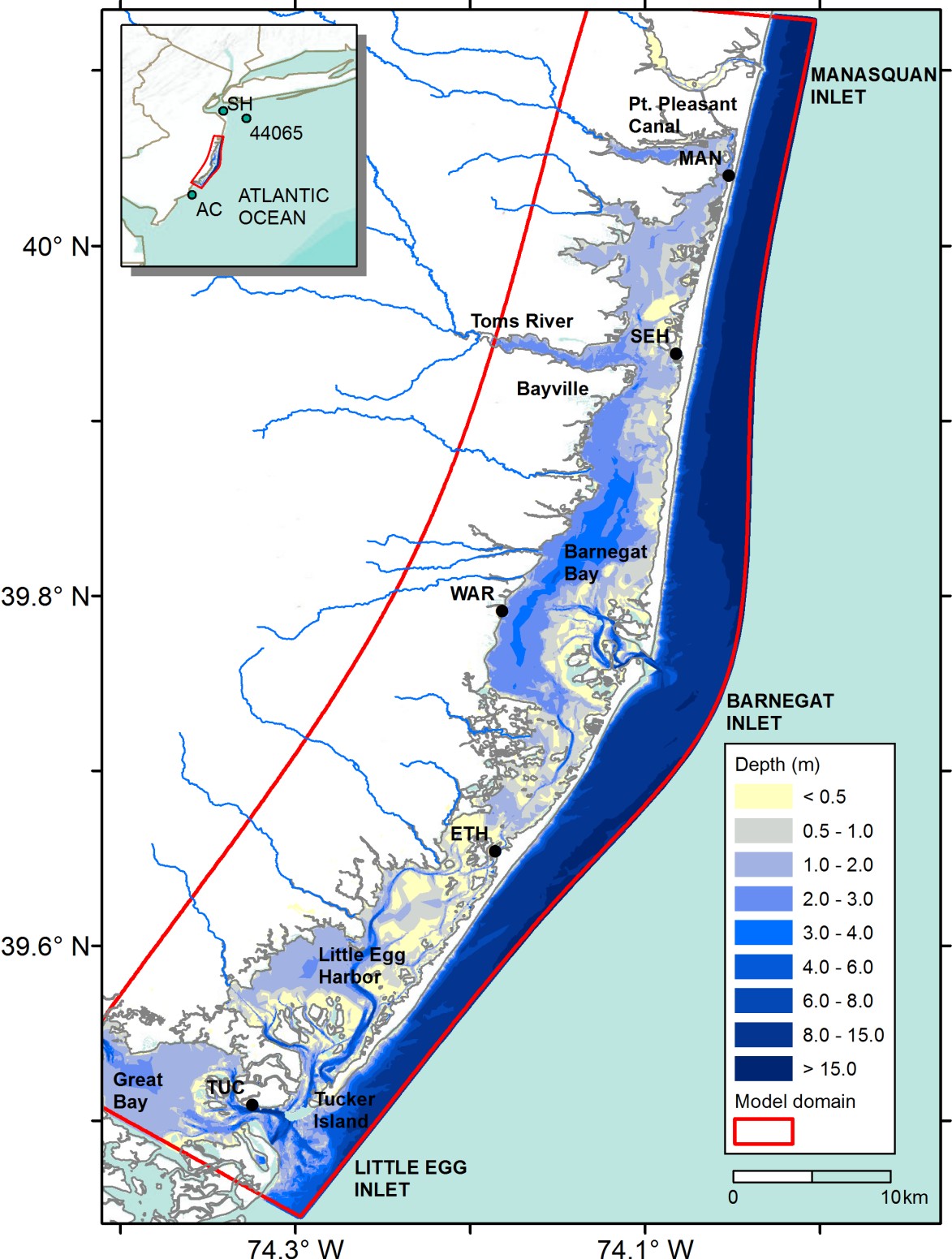

**Figure 1. Map of Barnegat Bay and Little Egg Harbor estuary showing the water level stations, bays, and inlets. The water level stations are: Tuckerton (TUC); East Thorofare (ETH); Waretown (WAR); Seaside Heights (SEH); and Mantoloking (MAN). Locations of offshore water level proxy stations and wind buoy are indicated in inset. COAWST model domain boundary is shown in red. Rte. 72 crosses the bay near ETH station and the breach that occurred during Hurricane Sandy was about a hundred meters away from MAN station, so they are not indicated in the map.**

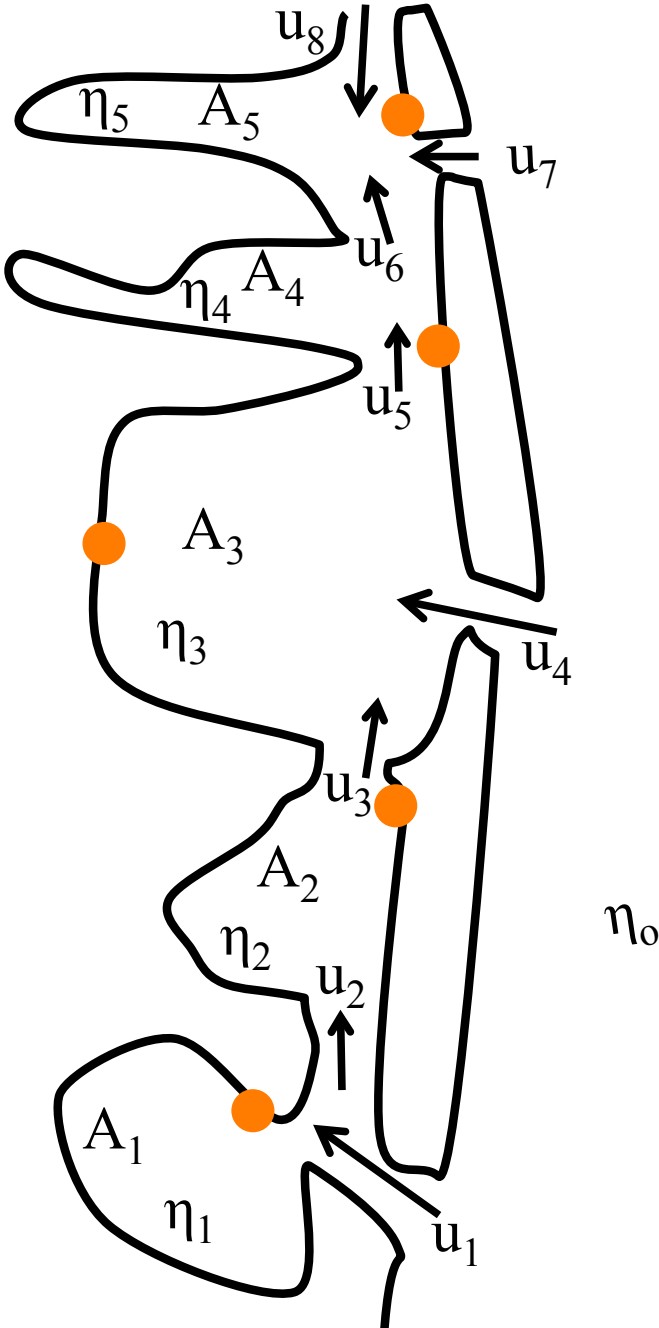

**Figure 2. Schematic diagram of the ocean-inlet-bay system: $A_j$ is the surface area of the bays; $\eta_j$ the sea level in the bays; $\eta_o$ the sea level in the ocean; and $u_j$ is the velocity through channel j. The correspondence with the real bay system includes**

areas from the bays (Great Bay, $A_1$; Little Egg Harbor, $A_2$; Barnegat Bay, $A_3$; Toms River sub-embayment, $A_4$; North of Mantoloking, $A_5$), flow through inlets (Little Egg Inlet, $u_1$; Barnegat Inlet, $u_4$; Point Pleasant Canal, $u_8$; and Mantoloking breach, $u_7$), and flow between bays (Tucker Island, $u_2$; Route 72, $u_3$; Bayville, $u_5$; Mantoloking, $u_6$). The location of the water level stations are indicated with dots and the names and specifications are in Figure 1 and Table 1.


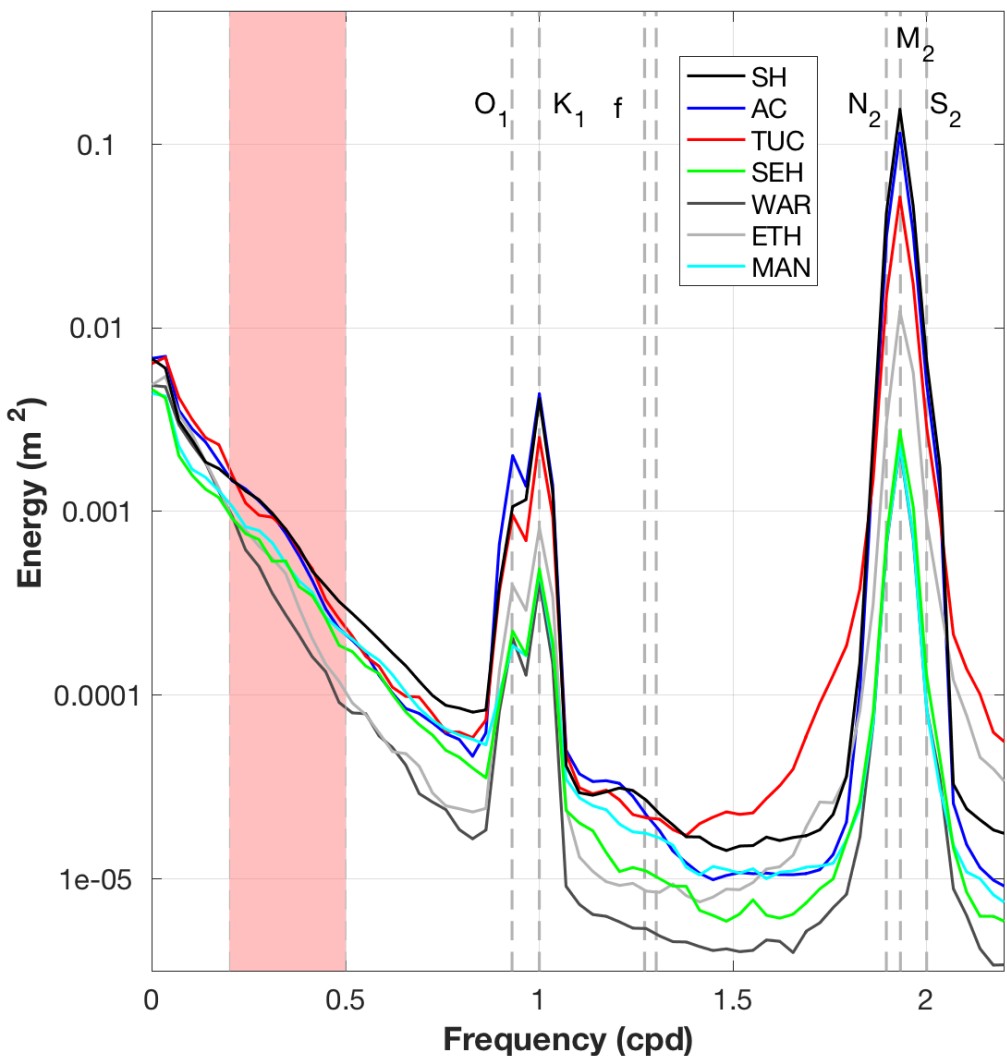

**Figure 3. Energy spectra at all stations computed using a Hanning 29-day window with over-lapping (50%) data segments. O₁, K₁, N₂, M₂, and S₂ label the principal tidal frequencies and *f* the inertial frequency. The vertical shaded area indicates the frequencies corresponding to the storm band (2-5 days). (cpd: cycles per day). See Table 1 for key to station abbreviations and Figure 1 for locations.**


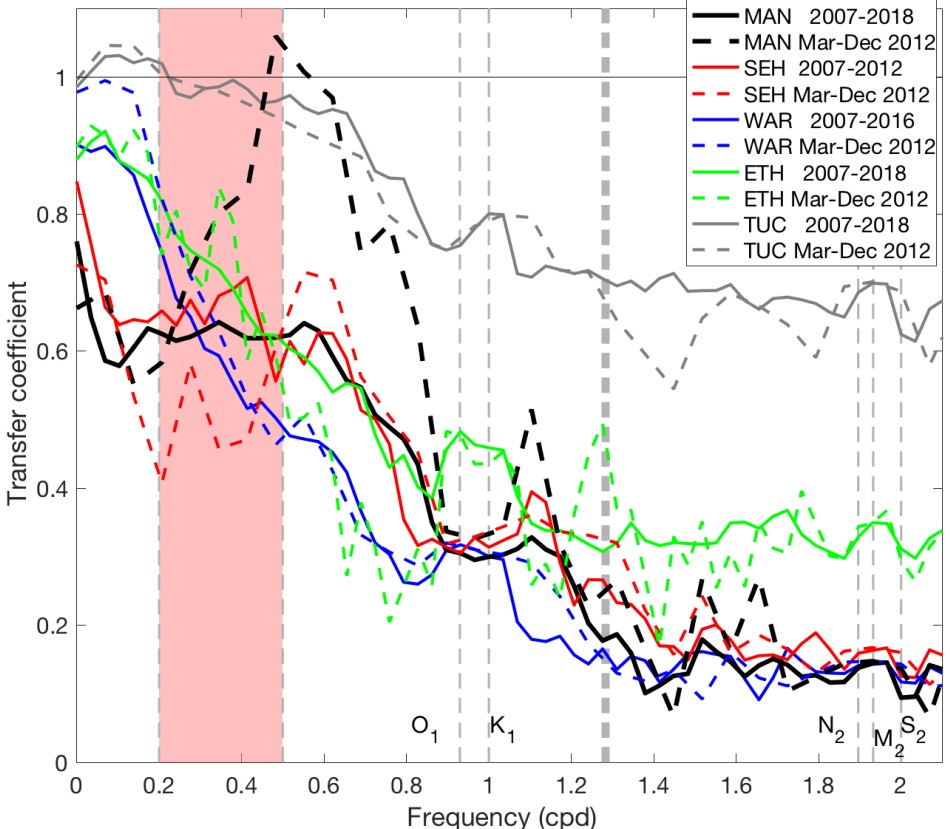

**Figure 4. Observed transfer from Atlantic City to 5 bay stations: Mantoloking (MAN), Seaside Heights (SEH), Waretown (WAR), East Thorofare (ETH), and Tuckerton (TUC). Solid lines indicate transfers for the entire available record at each station. Dashed lines represent observed transfers for the period March-December 2012, for which numerical model solutions were available. The vertical shaded area indicates the frequencies corresponding to the storm band (2-5 days).**


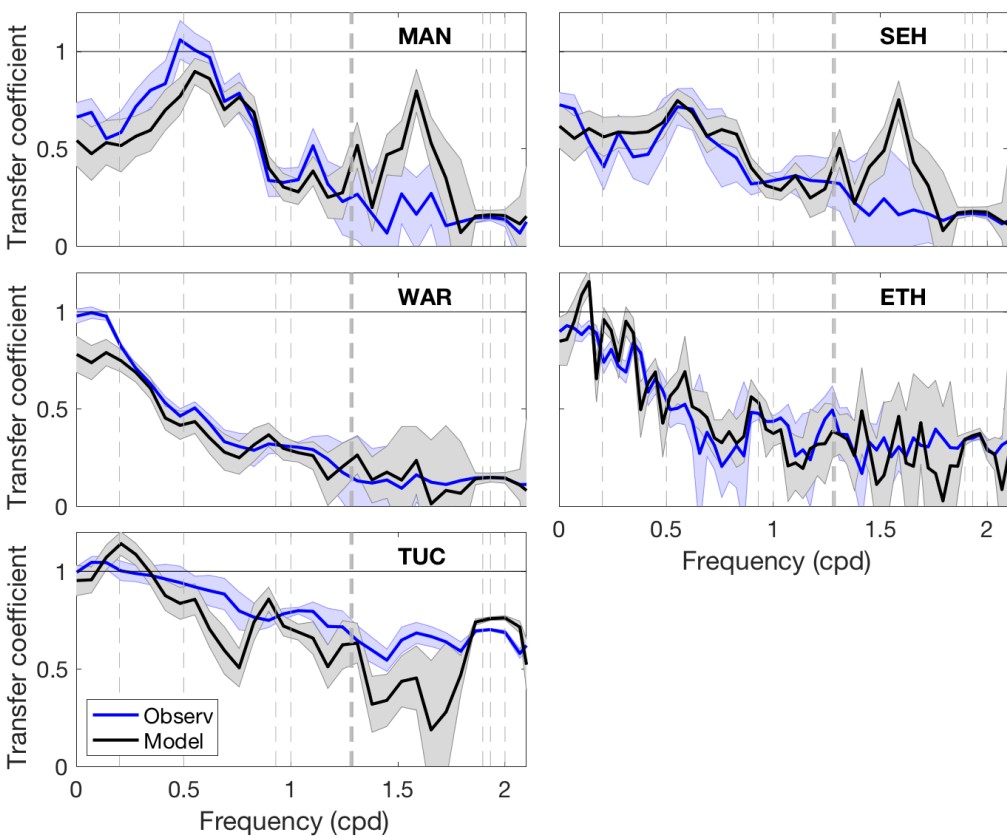

**Figure 5.** Comparison between observed (blue) and numerical model (black) transfers for the period when both are available (March-December 2012) at five bay stations. Uncertainty envelopes for the transfer coefficient (Bendat and Piersol, 1986) are provided for observed (light blue) and model (gray) estimates.

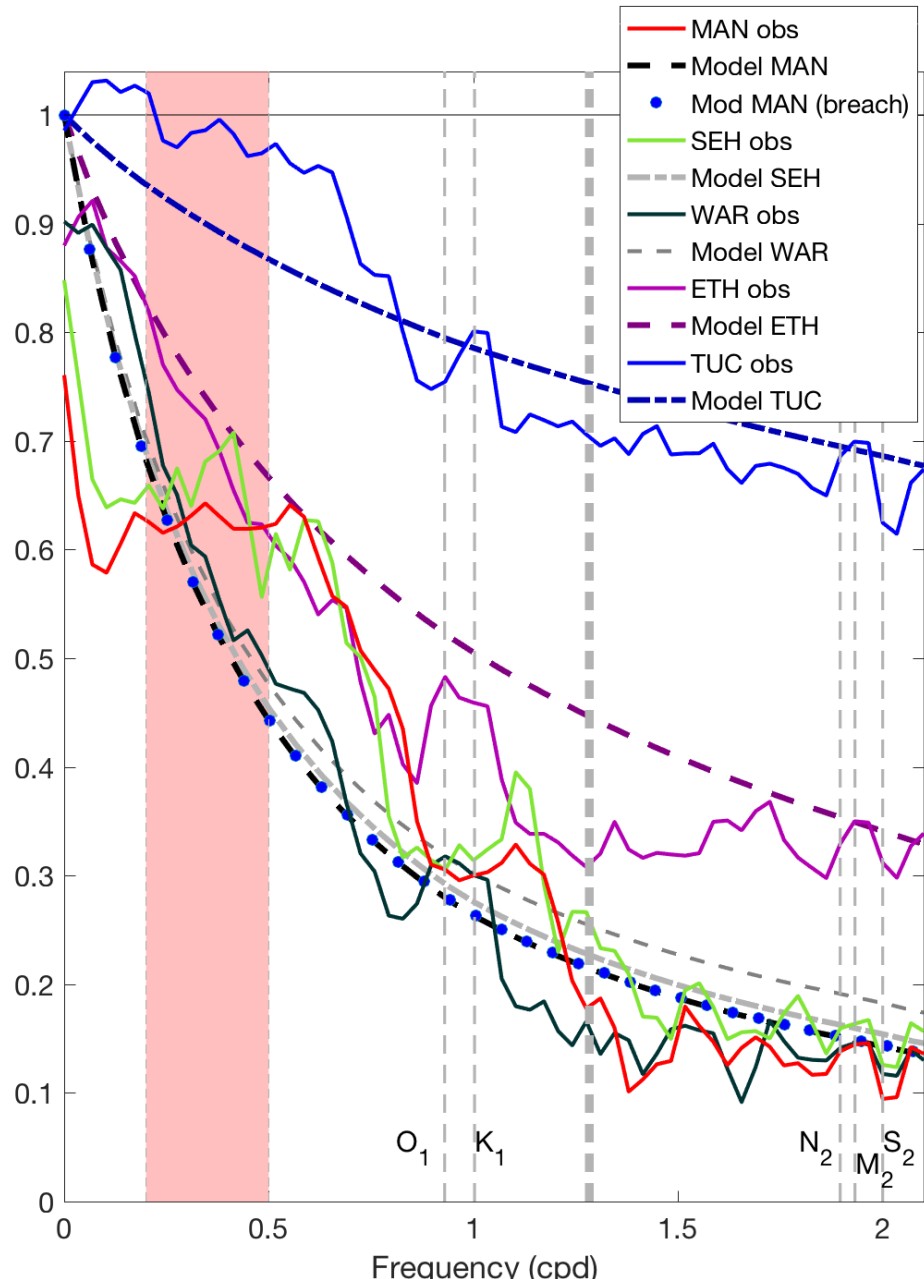

**Figure 6. Observed transfer for longest available record (solid lines) and best analytical model fit for each of the sub-embayments**
**(dashed lines). The vertical shaded area indicates the frequencies corresponding to the storm band (2-5 days).**

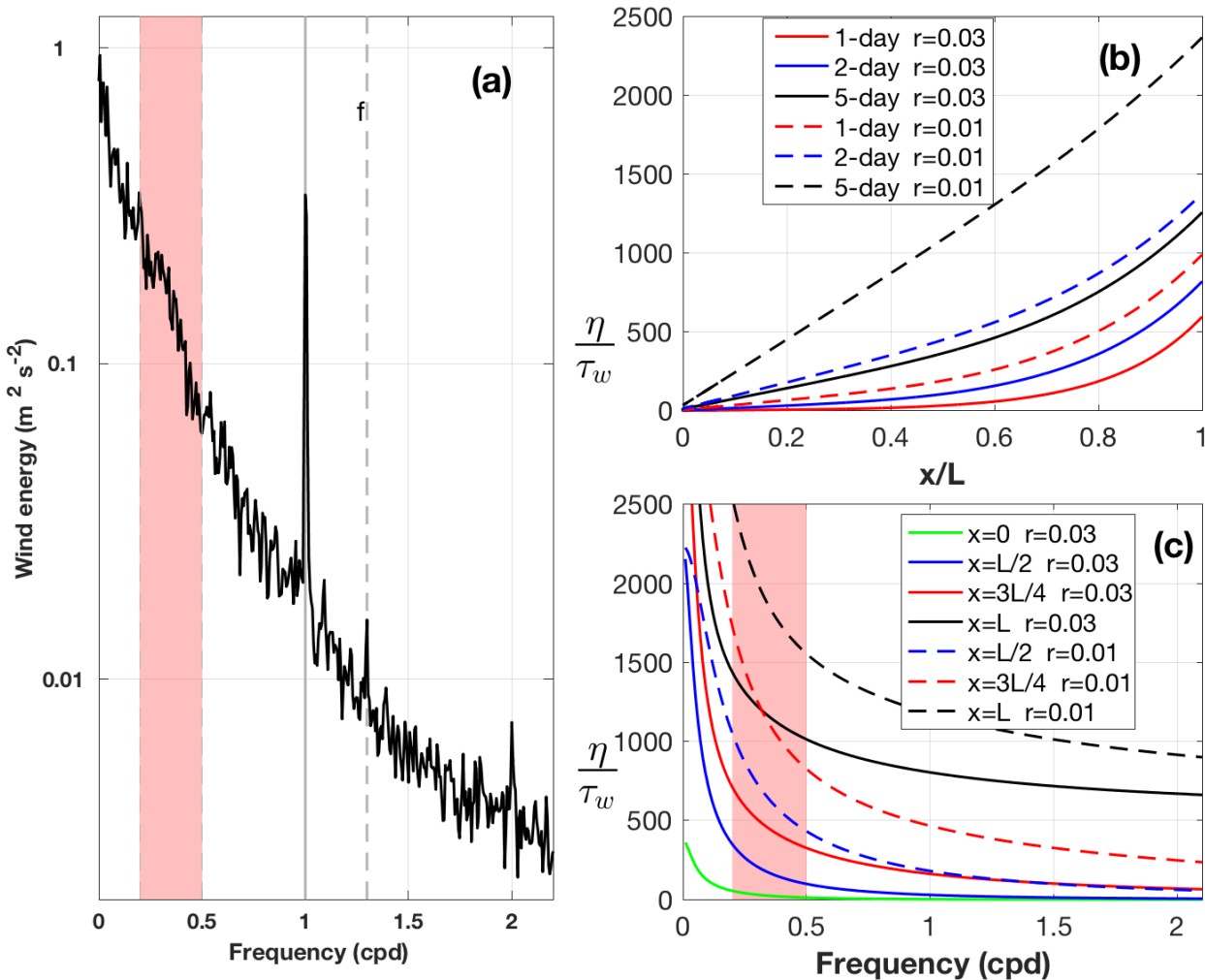

**Figure 7: (a) Wind speed spectra for the along-bay wind component for NDBC 44065 buoy (2008-2018). (b) Kinematic wind stress contribution to local water level in the bay expressed as $\eta/\tau_w$ (m$^{-1}$s$^2$) following the Wong and Moses-Hall (1998) formulation as a function of distance from the southern edge of the bay (x/L = 1 is the north edge of bay). (c) Kinematic wind stress contribution to water level as a function of frequency. The vertical shaded area indicates the frequencies corresponding to the storm band (2-5 days).**


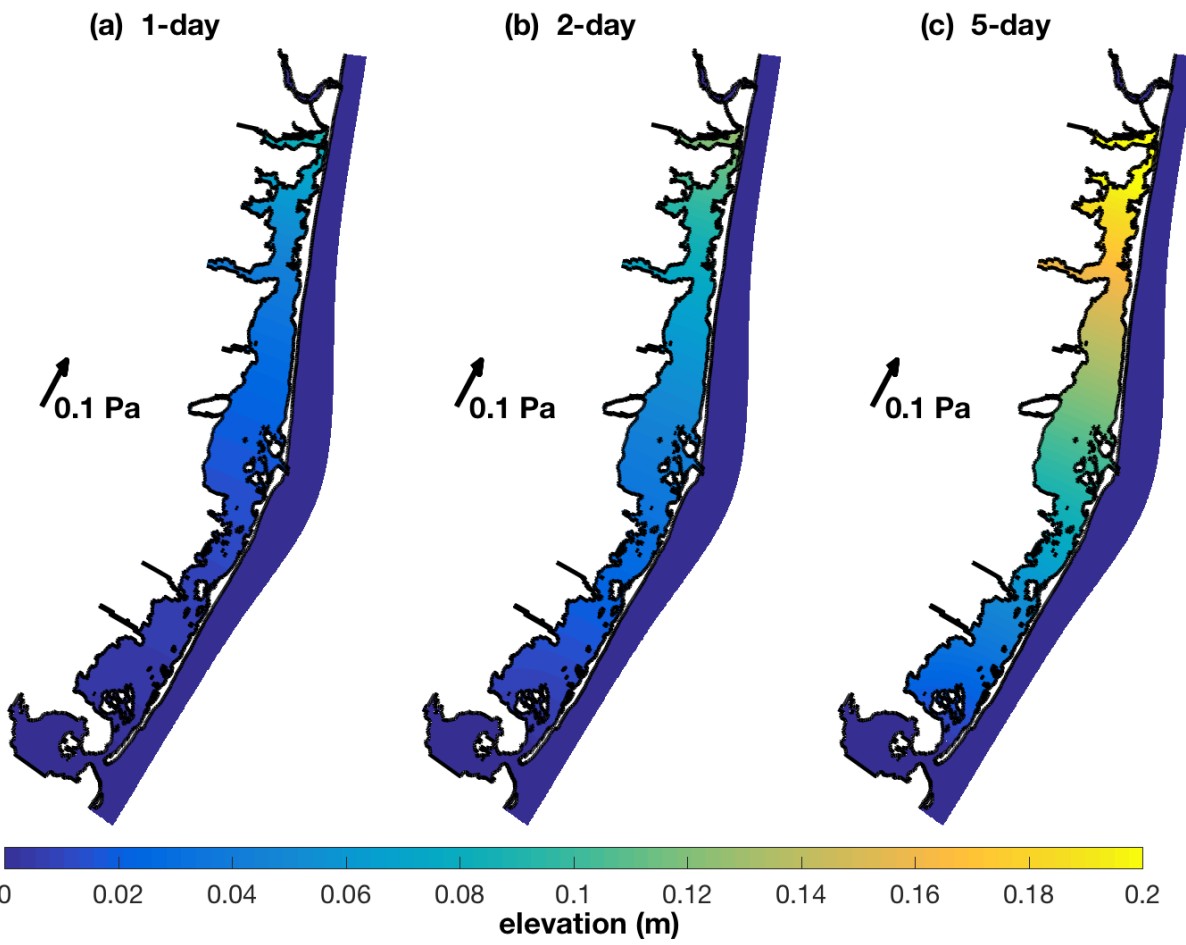

**Figure 8:** **Local wind setup inside the bay based on the Wong and Moses-Hall (1998) formulation for a wind stress of 0.1 Pa at**
**specific periods: (a) wind with a 1-day period (e.g., sea breeze); (b) 2-day wind; and (c) 5-day wind.**


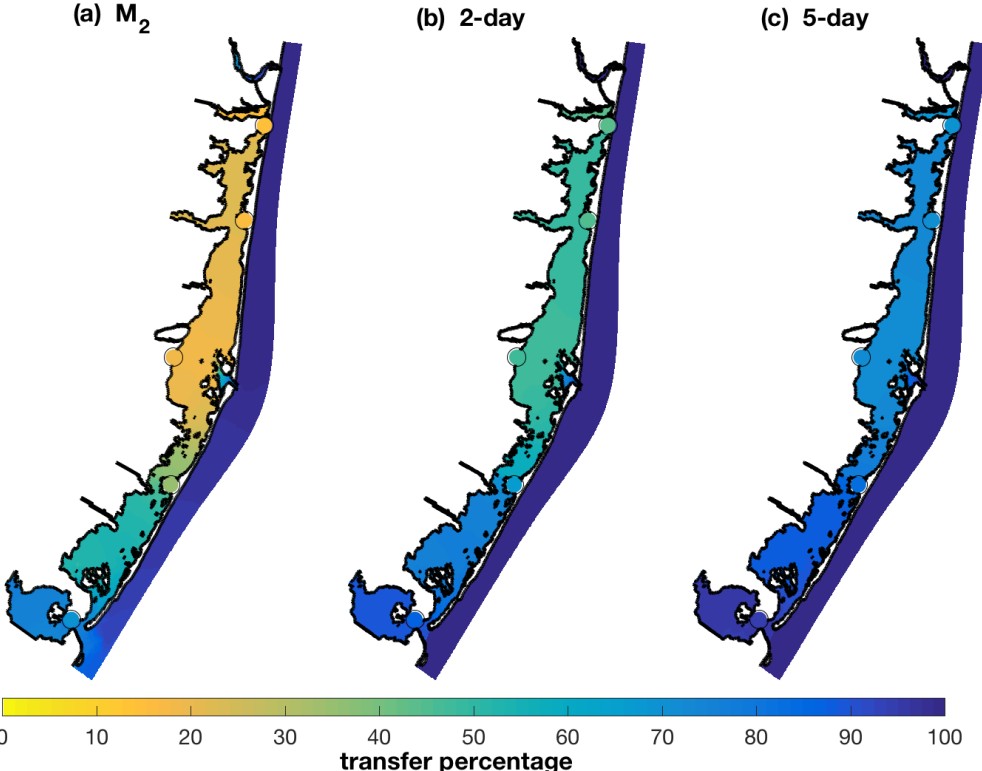

**Figure 9. Spatially variable transfer function (percentage) of offshore fluctuations transferred into the bays using Atlantic City as offshore proxy for three frequencies: (a) M₂ semidiurnal tide; (b) 2-day fluctuation in the storm band; and (c) 5-day fluctuation in the storm band. The filled circles represent the transfer estimate at each of the observed locations. Spatial pattern computed using the COAWST numerical solution.**


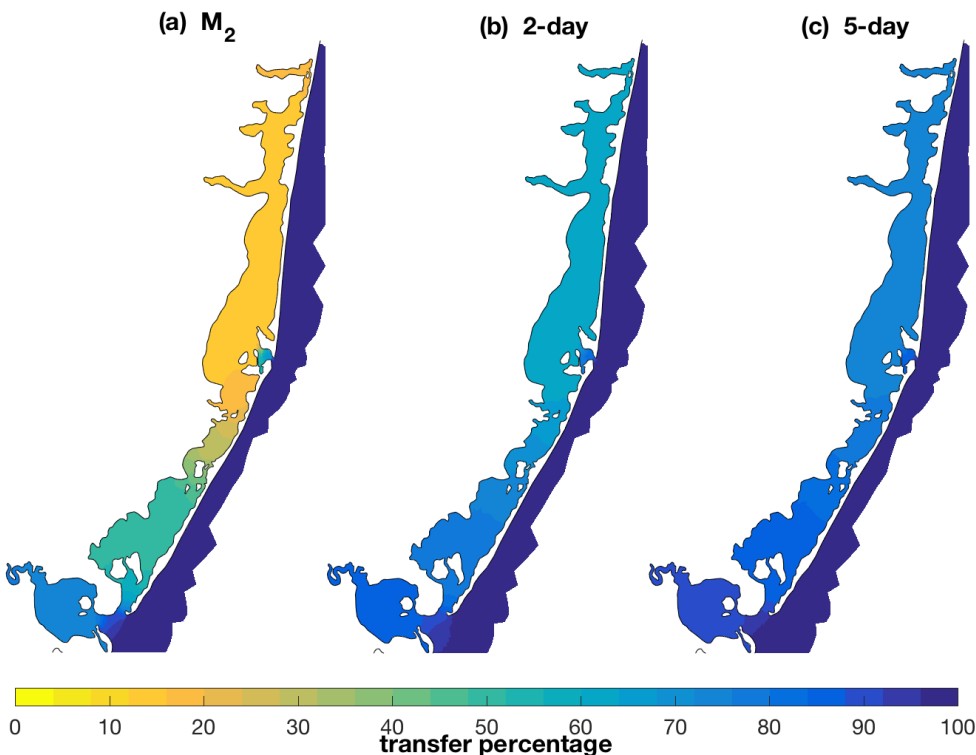

**Figure 10: Transfer estimate based on ADCIRC tidal database for three frequencies: (a) M2 semidiurnal tide; (b) 2-day fluctuation in the storm band; and (c) 5-day fluctuation in the storm band.**


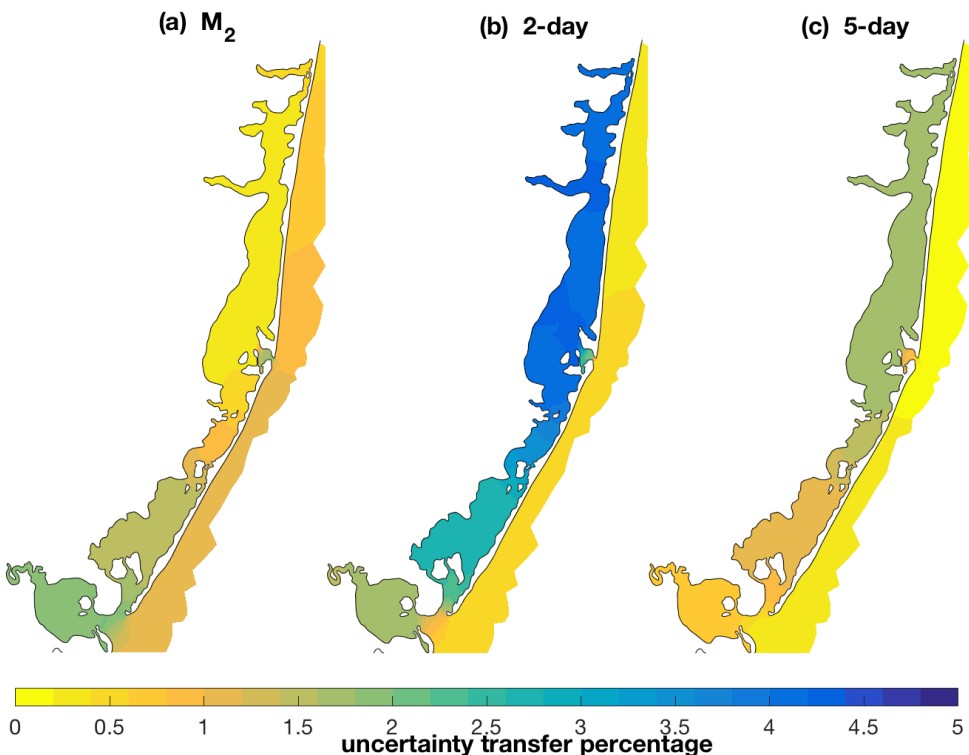

**Figure 11: Uncertainty in transfer estimate based on ADCIRC tidal database for three frequencies: (a) M2 semidiurnal tide; (b) 2-day fluctuation in the storm band; and (c) 5-day fluctuation in the storm band.**

**Table 1.** Sites used in water level analysis. Check Figure 1 for locations. National Oceanic and Atmospheric Administration (NOAA); U.S. Geological Survey (USGS). Information on instrumentation type, sampling and quality control methodologies for the USGS stations is available starting at https://waterdata.usgs.gov/nwis/inventory.

| Site name (Abbreviation) | Operator/ Site ID | Inlet/Bay | Available Period | Datum | Adjustment to NAVD88 |
|---|---|---|---|---|---|
| Sandy Hook, NJ (SH) | NOAA 8531680 | Offshore proxy | Jan 1910 – May 2018 | MSL | 0.15 m |
| Atlantic City, NJ (AC) | NOAA 8534720 | Offshore proxy | Aug 1911 – May 2018 | MSL | 0.12 m |
| Barnegat Bay at Mantoloking (MAN) | USGS 01408168 | Barnegat Bay | Oct 2007- May 2018 | NGVD29 | 0.34 m |
| Barnegat Bay at Seaside Heights (SEH) | USGS 01408750 | Barnegat Bay | Oct 2007- Oct 2012 | NGVD29 | 0.35 m |
| Barnegat Bay at Waretown (WAR) | USGS 01409110 | Barnegat Bay | Oct 2007- Dec 2016 | NGVD29 | 0.37 m |
| East Thorofare at Ship Bottom (ETH) | USGS 01409146 | Little Egg Harbor | Oct 2007- May 2018 | NGVD29 | 0.38 m |
| Little Egg Inlet near Tuckerton (TUC) | USGS 01409335 | Great Bay | Oct 2007- May 2018 | NGVD29 | 0.38 m |

**Table 2.** Sum of energy (m$^2$) in the different bands of the spectra computed for the period 2007-2018 (or longest available record) using a 29-day Hanning window with over-lapping (50%) data segments.

| Site | Low frequency | 2-5 days | 1-2 days | Diurnal tide | 0.5-1 days | Semidiurnal tide | High frequency |
|---|---|---|---|---|---|---|---|
| **SH** | 0.023 | 0.007 | 0.002 | 0.008 | 0.001 | 0.251 | 0.002 |
| **AC** | 0.025 | 0.007 | 0.001 | 0.010 | 0.001 | 0.187 | 0.001 |
| **TUC** | 0.025 | 0.006 | 0.001 | 0.005 | 0.001 | 0.090 | 0.001 |

| | | | | | | | |
|---|---|---|---|---|---|---|---|
| **ETH** | 0.020 | 0.004 | 0.001 | 0.002 | <0.001 | 0.023 | 0.001 |
| **WAR** | 0.019 | 0.003 | 0.001 | 0.001 | <0.001 | 0.004 | <0.001 |
| **SEH** | 0.015 | 0.004 | 0.001 | 0.001 | <0.001 | 0.005 | <0.001 |
| **MAN** | 0.015 | 0.005 | 0.001 | 0.001 | <0.001 | 0.004 | <0.001 |
