# Peer review of "Spatial distribution of water level impact to back-barrier bays"

_Natural Hazards and Earth System Sciences, 2018_

## Referee Comment (RC1) · Anonymous Referee #1 · 3 Jan 2019

The authors have proposed a novel approach to combine observed data and numerical model results for spatial characterization of water level transfer inside Barnegat Bay. They use dimensional characteristics of the bay to ensure this combination occurs in a physically consistent way. The idea is interesting and the manuscript is generally well-written, so I think it deserves publishing in NHESS after a major revision. Details are provided below:

Major:

- In page 5, with a harmonic assumption for water level and velocity, jumps into a giant equation (I wish there was an equation number I could refer to!). There is no way I can evaluate the robustness of approach, without knowing the exact steps and detailed assumption made here. I suggest, either providing enough details to enable proving

the accuracy of equations, or if there is not enough room in the main manuscript (which I think is the case) add supplementary materials and provide the detailed steps in that document.

- M2 is taken as a proxy for internal frictional effects (Page 7, Line 2). As far as I understand, overtides (i.e. M4) are better proxies for internal frictional effects. It's already been mentioned in the manuscript (Page 7, line 17) that remaining frequency bands exhibit smaller fluctuations, but their variability given forcing still contains useful information. Please, revise or justify this approach.

Minor:

- In Page 1, Line 33: there are many more recent citations to be cited here, including the revised version of this report in 2013. Also, see the followings for example:

* Rahmstorf (2017) Rising hazard of storm-surge flooding, PNAS, https://doi.org/10.1073/pnas.1715895114

* Wahl et al. (2017) Understanding extreme sea levels for broad-scale coastal impact and adaptation analysis, Nature Communications volume 8, Article number: 16075.

- In Page 2, Line 32: The following paper may be cited to define the term nuisance flooding for interested readers. * Moftakhari et al. (2018) What is nuisance flooding? Defining and monitoring an emerging challenge, Water Resources Research 54 (7), 4218-4227.

- In Page 3, lines 3-4: cite more recent literature, as you are pointing to the gap and we need to make sure the gap has not been filled since 2000.

- Please use different notation in harmonic assumption for amplitude and actual fluctuating variable (i.e. saying u=u*exp(iwt) is confusing)

- In Page 5, Line 3: please be specific what kind relationship would be described by phi parameter (linear? nonlinear?...)

Good luck,

---

## Referee Comment (RC2) · Anonymous Referee #2 · 7 Feb 2019

**GENERAL COMMENTS**

The paper is impressive and can be influential, with some excellent ideas and its broad perspective based on observations, detailed numerical modeling, analytical modeling, and also possible extension nationwide using an ADCIRC tide constituent database. However, the analytical developments are dense and could be explained better for a less technical reader. Also, and most importantly, the discussion of potential use with ADCIRC tide modeling results datasets in storm hazard assessment needs work. I believe a major shortcoming there is the neglect of local wind setup in storms. Back-bays can have a wide range of inlet sizes, bay area, and often have shallow water depths, and as a result, can have an important role for local wind setup in storms. The paper can acknowledge this, if the authors agree, and it will be a stronger paper. As a

result, I recommend major revision.

SPECIFIC COMMENTS

ABSTRACT

A minor comment – text refers to "Inlet geometry and bathymetry" as being important in semi-enclosed bays - isn't bay area also important?

The abstract says storm transfers were from 70-100% but I see ~50% in some cases-eg MAN at 5-day period, WAR at 2-days. So this should be revised to 50-100%.

The last several sentences of the abstract don't seem very consistent with the paper's discussion- differing topics are discussed. Also where is mention of the ADCIRC based transfer estimates?

INTRO vs METHODS

A part of the intro's literature review says that wind controls backbay currents (Garvine 1985). But in the methods, the approach uses tidal current M2 are a proxy for bed friction.

Section 4.2, p. 6 – on wind's influence – analytical model – instead of saying "angular frequency" would it be more clear to say "cyclic frequency"? The figures show "cycles/day" and "angular frequency" just is a little confusing to me. It isn't measured as an angle (degrees), it's measured by cycles. Here, you also might refer to the two different stresses as "dynamic stress" (tau_s I believe) and "kinematic stress" (tau_w), as well as in the figure caption. It would have helped me a little in understanding the figure's values (values of order 1000) as I typically think in terms of dynamic stress in Pascals.

I am a bit confused about why tau is written as a function of omega (parenthetically) here so it is probably a good thing to explain things more. I see wind stress and frequency as being independent variables.

[Figure]

Also, does the denominator really include cos(Lx) here? I see L as being on the order of 10000m and x being from 0 to 10000 (meters).

Section 6.1- aspects regarding sandy don't seem very useful here – see points below on local wind setup

"this far" – requires a minor revision to "thus far" I believe

Section 6.2

A very interesting idea and impressive analysis and results

Section 6.3

Seems to be a fairly ingenious approach! are all US backbays really available and well-resolved in the ADCIRC tide data? In this case there is a large inlet that controls results, but I expect there are tougher cases.

A little more elaboration or demonstration here might be useful – it's the final landing point of the paper and seems like it could be helpful to illustrate this potential application with more detail.

The claim in the paper seems to be that local wind setup is small and negligible for storms, relative to transfer of offshore surge. The maximum wind setup mentioned is only 20cm (p10, line16). This all seems surprising to me, as I have learned that (large, shallow) backbay surge is often strongly inflenced by local winds.

A challenge: the wind setup for the north winds prior to Sandy's landfall was studied, when winds blew water toward (fortunately!) the main inlet. How about computing and comparing setup for when the wind turned around and blew from the south after landfall, toward the nearly dead-end northern end of the bay? If we are interested in hazards, then this was the primary backbay damage-causing period of the event. I believe the local wind setup became quite large and abrupt at that time.

During Sandy, water levels at Mantoloking rose about 2.5m in 12 hours when the wind

rotated to come from the east and then south, reaching a maximum that was very close to the open ocean or Little Egg values (perhaps 30cm lower) (USGS station 01408168).

Does the analytical approach capture the effect of this large wind setup? If not, does this issue show that local wind setup can be a challenge for using the ADCIRC tide data to estimate storm hazards? Or, is Sandy too unusual of a case, in which case a nor'easter might be a better discussion point for the paper?

Using a back-of-the-envelope computation with an admittedly simpler, but well-established method to compute the wind setup for a 20m/s wind, 50km fetch, 2m deep backbay, if fully developed, is 2m (the Zuider-Zee equation, or similarly, making the computation using a steady state vertically averaged momentum budget - Pugh and Woodworth, 2014, p. 156 in Section 7.3 on Storm Surges). It takes only a matter of hours to fully develop. I agree on the 1Pa wind stress for Sandy- this is reasonable.

U=20m/s eg post-landfall Sandy

depth d=2m

fetch F=50000m

setup S= 0.000002 * F * U^2 /g /d = 0.000002 * 50000* 400/9.8/2 (Zuider-Zee equation)

S = 2m setup

I believe "L" in the analytical wind formulation is the basin length (for each of several small basins). I am computing the wind setup for a 50km long backbay, so perhaps using a longer fetch. But I believe this is appropriate as they are really not disconnected and the ~40km of the northern half of the model domain is strongly connected (not divided up into separate bays).

Sandy's surge might be viewed as having a "slow" 1-m surge with timescale of 3 days, plus a fast 1-m surge with a timescale of 1 day. I think for either case the ADCIRC-based transfer results in this paper suggest reduced transfer (maybe 60%; figure 10)

for Mantoloking. In contrast, Sandy, the worst extreme event, shows that local wind effects lead to a similar surge there as seen offshore (perhaps 90% transfer). The transfer uncertainty estimates in Figure 11 (eg 4% in the 2-day storm band?) aren't evaluating this wind setup contribution, so aren't worth much.

I think this local wind effect pushing water into the northern end of the bay is what causes the unexplained high transfer at Mantoloking (Figures 4-5) in the 2-day storm band. The model captures it because it includes wind forcing (and Sandy), but a tide-only model will not.

To conclude, I think the method presented would, for storms, often be low-biased for peak water level risk estimates, due to local wind setup. I suggest proceeding very carefully and validating storm hazard estimates, using observational data.... or evaluating case studies carefully to determine whether local wind setup corrections or larger uncertainties can be added for the storm driven flood hazards. Underestimating storm-driven flood risk is worse than not estimating it at all.

REFERENCE Pugh and Woodworth, 2014, Sea Level Science, Second Edition, Cambridge University Press, Cambridge, UK.

---

## Author Response (AR1)

**Response to Reviewers, comments in plain text, response in bold**

5 The authors have proposed a novel approach to combine observed data and numerical model results for spatial characterization of water level transfer inside Barnegat Bay. They use dimensional characteristics of the bay to ensure this combination occurs in a physically consistent way. The idea is interesting and the manuscript is generally well-written, so I think it deserves publishing in NHESS after a major revision. Details are provided below:

*Major:*

10 - In page 5, with a harmonic assumption for water level and velocity, jumps into a giant equation (I wish there was an equation number I could refer to!). There is no way I can evaluate the robustness of approach, without knowing the exact steps and detailed assumption made here. I suggest, either providing enough details to enable proving the accuracy of equations, or if there is not enough room in the main manuscript (which I think is the case) add supplementary materials and provide the detailed steps in that document.

15 **The development of the equations is now provided in an appendix (Appendix A). Originally, we decided to exclude them as they take up a lot of space. Also, the equation numbers have been added to the text.**

- M2 is taken as a proxy for internal frictional effects (Page 7, Line 2). As far as I understand, overtides (i.e. M4) are better proxies for internal frictional effects. It's already been mentioned in the manuscript (Page 7, line 17) that remaining frequency bands exhibit smaller fluctuations, but their variability given forcing still contains useful information. Please,

20 revise or justify this approach.

**The $M_4$ tidal constituent is generated as a non-linear response to $M_2$ tidal forcing. The magnitude of the $M_4$ is partially a result of the energy loss from $M_2$ through friction but also associated with other nonlinearities. In general, the $M_4$ is a result of asymmetries in the duration of ebb and flood. There is no external solar or lunar forcing at the $M_4$ frequency. Thus, as the Reviewer mentions, the $M_4$ and other overtides might be related to internal frictional**

25 **effects in terms of where the $M_4$ is generated. In fact, the tidal constituent more associated with frictional generation is $M_6$ rather than $M_4$. The issue is that the $M_4$ also propagates as a normal tidal wave and the magnitude at a specific location can be the result of either local generation or propagation and it will also be subject to attenuation by friction. The spatial changes in the $M_2$ tidal constituent inside the bays are a direct consequence of the frictional dissipation of tidal energy through friction (Redfield, 1980). Therefore, the changes in $M_2$ amplitude are a better**

30 **metric for the frictional effects inside the bay.**

- In Page 1, Line 33: there are many more recent citations to be cited here, including the revised version of this report in 2013. Also, see the followings for example:

* Rahmstorf (2017) Rising hazard of storm-surge flooding, PNAS, https://doi.org/10.1073/pnas.1715895114

5  * Wahl et al. (2017) Understanding extreme sea levels for broad-scale coastal impact and adaptation analysis, Nature Communications volume 8, Article number: 16075.

**We have added the references mentioned by the Reviewer.**

**Ln 30-32: "Both hurricanes and winter storms affect coastal populations, infrastructure, and natural resources along the coastal bays of the United States (Nicholls et al., 2007, 2014; Rahmstorf, 2017; Wahl et al., 2017)."**

10  - In Page 2, Line 32: The following paper may be cited to define the term nuisance flooding for interested readers. * Moftakhari et al. (2018) What is nuisance flooding? Defining and monitoring an emerging challenge, Water Resources Research 54 (7), 4218-4227.

**The reference has been added to the text.**

**Ln 63-65: "The method will be useful for coastal hazard assessment assisting in the management of nuisance flooding**

15  **(Moftakhari et al., 2018) and providing spatial differences in vulnerability to perigean spring tides (king tides) and planning for flooding in response to storms of different durations."**

- In Page 3, lines 3-4: cite more recent literature, as you are pointing to the gap and we need to make sure the gap has not been filled since 2000.

**We have added recent references to the text to show the continuing coastal focus. For instance:**

20  - **Neumann, B., Vafeidis, A.T., Zimmermann, J. and Nicholls, R.J.: Future coastal population growth and exposure to sea-level rise and coastal flooding-a global assessment. PloS one, 10(3), p.e0118571, 2015.**
   - **Vitousek, S., Barnard, P.L., Fletcher, C.H., Frazer, N., Erikson, L. and Storlazzi, C.D.: Doubling of coastal flooding frequency within decades due to sea-level rise. Scientific reports, 7(1), p.1399, 2017.**

**Ln 34-38: "While flooding in the mainland side of back-barrier bays has severe socio-economic implications, most of**

25  **the coastal hazard evaluations (Gornitz et al., 1994; Thieler and Hammar-Klose, 1999; Klein and Nicholls, 1999; Kunreuther et al., 2000; Neumann et al., 2015; Vitousek et al., 2017) have focused in open-coast areas. Vulnerability evaluation of coastal areas around back-barrier bays requires extensive knowledge of the main hazard sources and their physical controls."**

- Please use different notation in harmonic assumption for amplitude and actual fluctuating variable (i.e. saying

30  u=u*exp(iwt) is confusing)

**We have improved the notation accordingly and differentiated between the amplitude and the fluctuating variable. For instance, Ln 120-121: "Assuming $\eta = \tilde{\eta}e^{i\omega t}$ and $u = \tilde{u}e^{i\omega t}$, where $\tilde{\eta}$ and $\tilde{u}$, represent the magnitude of the water level and velocity oscillations, respectively."**

- In Page 5, Line 3: please be specific what kind relationship would be described by phi parameter (linear? nonlinear?...)

The relationship that the parameter phi represents is a linear relationship that is consistent with the linear equation described in the text. We have clarified this characteristic in the text.

Ln 117-119: "$\phi_{LEI}$, $\phi_{BI}$, $\phi_{breach}$, $\phi_{PPC}$ are the linear frequency-dependent relationships between the water levels at offshore proxy stations (Sandy Hook or Atlantic City) and the water level just offshore of Little Egg Inlet, Barnegat Inlet, the breach at Mantoloking caused by Sandy, and Point Pleasant Canal."

Reference:

Redfield, A.C., 1980. "The tides of the waters of New England and New York", 109 pp., doi:10.1575/1912/1136, https://hdl.handle.net/1912/1136

The paper is impressive and can be influential, with some excellent ideas and its broad perspective based on observations, detailed numerical modeling, analytical modeling, and also possible extension nationwide using an ADCIRC tide constituent database. However, the analytical developments are dense and could be explained better for a less technical reader. Also, and most importantly, the discussion of potential use with ADCIRC tide modeling results datasets in storm hazard assessment needs work. I believe a major shortcoming there is the neglect of local wind setup in storms. Back-bays can have a wide range of inlet sizes, bay area, and often have shallow water depths, and as a result, can have an important role for local wind setup in storms. The paper can acknowledge this, if the authors agree, and it will be a stronger paper. As a result, I recommend major revision.

**The reviewer is completely correct. There was an error in the calculations of the local wind setup. The formulas were fine but there was an error on the implementation of the formulas. After the error was corrected, the effect of the wind was substantially bigger (5-10 times). The new results have been included and the paper makes a much better argument about the importance of wind setup thanks to the suggestions made by the reviewer.**

**We included the wind effects on the bay in two sections of the paper. Section 4.2 and 4.3 include the methodology, while Section 5.2 includes the results of the wind effect on the bay. We have added the wind effects in the discussion of the ADCIRC results as the Reviewer suggested.**

**Ln 243-251: "The resulting effect of the wind setup (or set-down) was small (less than 0.1 m with an along-bay wind stress of 0.1 Pa) for most of the domain (Figure 8). The estimate assumed a linear friction of the same magnitude as in Section 5.1 (r=0.021 m/s). Under persistent wind stress of 0.1 Pa (about 8 m/s wind speed) in the along-bay direction, the resulting setups varied depending on the frequency considered. Setup magnitudes over 0.2 m were estimated for the 5-day period wind (Figure 8c), while under half of that magnitude was achieved for the 2-day persistent wind (Figure 8b), and much smaller water level setup (peak smaller than 0.1 m) was estimated for the sea breeze (Figure 8a). During extreme events like Hurricane Sandy, under intense wind stress, two additional effects should be considered: the depth of the bay increases by the transfer of offshore surge resulting in altered setup response (Section 4.2), and the frictional effect is enhanced (a larger linear friction would be needed) by the presence of wave-induced roughness."**

**Ln 273-282: "The wind setup effect inside the bay due to local wind can also be estimated for Hurricane Sandy using the approach in Section 4.2. Maximum wind stress during the storm was about 1 Pa. To obtain a maximum effect (worst-case scenario) the wind was assumed to be persistently in the along-bay direction and that maximum stress was maintained for the duration of the storm. The maximum resulting water level considering the Wong and Moses-Hall method is linear with regard to wind stress magnitude (Figure 7b) and would have been 10 times larger than the setup in Figure 8b. The maximum wind setup would have been between 1 and 2 m, which was of the same order of magnitude as the surge produced from offshore sources. The cross-bay contribution to the wind setup during Sandy was comparatively small as wind direction was predominantly along-bay. Surge estimates from simple analytical formulations (State Committee for the Zuiderzee, 1926; Pugh, 1987) that do not consider storm duration produce similar magnitude results and are also dependent on the frictional response of the bay."**

**Ln 314-319: "The effect of local wind setup will also need to be added to the ADCIRC-based estimate, especially during severe storms. The approach discussed in Section 5.2 or even a simpler surge calculation (e.g., from the steady state vertically averaged momentum equations, as in Pugh (1987), from the traditional report of the State Committee for the Zuiderzee (1926), or the updated frequency domain equivalent from Reef et al., 2018) could be used and the resulting**

elevation could be added to the offshore transfer estimate obtain based on the ADCIRC tides. Thus, the production of bay water level predictions will require accurate wind forecast products."

ABSTRACT

A minor comment – text refers to "Inlet geometry and bathymetry" as being important in semi-enclosed bays - isn't bay area also important?

**Bay geometry was included two sentences before. We have added bay area to that sentence so that it reads: "Bay area and inlet geometry and bathymetry primarily regulate the magnitude of the transfer between open ocean and bay."**

The abstract says storm transfers were from 70-100% but I see $\sim$50% in some cases- eg MAN at 5-day period, WAR at 2-days. So this should be revised to 50-100%.

**Agreed. It has been revised according to the Reviewer's suggestion.**

**Ln 16-18: "Model water level transfers match observed values at locations inside the Bay in the storm frequency band (transfers ranging from 50-100%) and tidal frequencies (10-55%)."**

The last several sentences of the abstract don't seem very consistent with the paper's discussion- differing topics are discussed. Also where is mention of the ADCIRC based transfer estimates?

**The second to last sentence has been removed and an alternative has been added that explains the ADCIRC-based approach potential for expansion to other areas. It reads:**

**Ln 21-22: "An extension of the methodology that takes advantage of the ADCIRC tidal database for the east coast of the United States allows for the expansion of the approach to other bay systems."**

INTRO vs METHODS

A part of the intro's literature review says that wind controls backbay currents (Garvine 1985). But in the methods, the approach uses tidal current M2 are a proxy for bed friction.

**While wind has a large influence in back-bay current variability, the character of the wind response is less predictable. The total current is mainly a result of the combined effect of wind and tides. In the absence of wind, the tides will always be there. Estimating bottom friction based only on the M2 tidal current might underestimate friction in cases of large wind currents. In reality, the linear friction estimate is based on the match of the M2 tidal amplitude in the numerical model and the observations and as such it includes any components of the velocity that has affected the observations and resulted in the measured water level signals. This point has been clarified in the text by adding:**

**Ln 165-170: "As most of the water level variability in the bay is associated with the M2 semidiurnal tidal constituent (Figure 3) and the distribution of the tide has been properly validated in the numerical simulations (Defne and Ganju, 2015), we can take the spatial distribution of the M2 tidal amplitude as a proxy for the internal frictional effects in the bay. Bottom friction caused by both wind driven and tidal effects is considered in the numerical simulations. By adjusting the water level based on the numerical M2 spatial distribution, we are approximating the complete frictional characteristics of the bay."**

Section 4.2, p. 6 – on wind's influence – analytical model – instead of saying "angular frequency" would it be more clear to say "cyclic frequency"? The figures show "cycles/day" and "angular frequency" just is a little confusing to me. It isn't measured as an angle (degrees), it's measured by cycles.

**We have included the suggestion to avoid confusion. It now reads, Ln 145: "… ω is the cyclic (angular) frequency".**

Here, you also might refer to the two different stresses as "dynamic stress" (tau_s I believe) and "kinematic stress" (tau_w), as well as in the figure caption. It would have helped me a little in understanding the figure's values (values of order 1000) as I typically think in terms of dynamic stress in Pascals.

**We have changed the text accordingly.**

**Ln 142-144: "$\tau_s$ and $\tau_b$ are the surface and bottom dynamic stresses, respectively; and $\rho_0$ is the water density. $\tau_w = {}^{\tau_s}\!/\rho_0$ is the spatially invariant kinematic wind stress and $r$ is a linearized bottom friction."**

**Ln 148-149: "$\widetilde{\tau_w}(\omega)$ represents the magnitude of the kinematic wind stress that results in water level fluctuations at a specific frequency."**

I am a bit confused about why tau is written as a function of omega (parenthetically) here so it is probably a good thing to explain things more. I see wind stress and frequency as being independent variables.

**The wind stress represented here is the magnitude of the wind oscillations that result in water level fluctuations at a specific frequency.**

Also, does the denominator really include cos(Lx) here? I see L as being on the order of 10000m and x being from 0 to 10000 (meters).

**That is a typo. The equation should read cos(kL) instead. It has been corrected in the text.**

Section 6.1- aspects regarding sandy don't seem very useful here – see points below on local wind setup
"this far" – requires a minor revision to "thus far" I believe Section 6.2 **It has been corrected**

A very interesting idea and impressive analysis and results Section 6.3 **Thanks. We believe it has the potential to help in the prediction of bay water level hazards.**

Seems to be a fairly ingenious approach! are all US backbays really available and well-resolved in the ADCIRC tide data? In this case there is a large inlet that controls results, but I expect there are tougher cases.

**Clearly not all back-barrier bays are well resolved in the tidal database, as some are relatively small. The ADCIRC model run provides resolutions down to around 50m in the latest database. The ADCIRC group keeps improving the resolution of their products. It is likely that their solutions will soon improve their quality and resolution, thus having the potential for providing even better back-bay response. The process of evaluating the quality of the current approach is underway, but so far it exhibits great potential.**

A little more elaboration or demonstration here might be useful – it's the final landing point of the paper and seems like it could be helpful to illustrate this potential application with more detail.

**The paper introduces the methodology based on the ADCIRC tides for the creation of offshore transfer estimates. The potential use for water level prediction in bays requires very careful calibration and skill assessment. The transfer maps represent an approximation to the average bay response to offshore forcing and specific events can depart significantly from the average response. Effects like overtopping of the barrier island, wave setup and runup, intense local wind setup, changes in frictional characteristics during a storm cannot be adequately predicted with the approach and need to be carefully included in the application to water level forecast.**

The claim in the paper seems to be that local wind setup is small and negligible for storms, relative to transfer of offshore surge. The maximum wind setup mentioned is only 20cm (p10, line16). This all seems surprising to me, as I have learned that (large, shallow) backbay surge is often strongly influenced by local winds.

**The reviewer is completely correct. There was an error in the calculations of the local wind setup. The formulas were fine (except from the typo mentioned above). The error was on the implementation of the formulas. After the error was corrected, the effect of the wind was much bigger (5-10 times). The maximum wind setup is now between 1 and 2 m. The text was changed accordingly. The new Figure 8 shows the same pattern, but the magnitude is much bigger:**

[Figure]

**New Figure 8: Local wind setup inside the bay based on the Wong and Moses-Hall (1998) formulation for a wind stress of 0.1 Pa at specific periods: (a) wind with a 1-day period (e.g., sea breeze); (b) 2-day wind; and (c) 5-day wind.**

**The text has been modified to read:**
**Ln 245-248: "Under persistent wind stress of 0.1 Pa (about 8 m/s wind speed) in the along-bay direction, the resulting setups varied depending on the frequency considered. Setup magnitudes over 0.2 m were estimated for the 5-day period wind (Figure 8c), while under half of that magnitude was achieved for the 2-day persistent wind (Figure 8b), and much smaller water level setup (peak smaller than 0.1 m) was estimated for the sea breeze (Figure 8a)."**

A challenge: the wind setup for the north winds prior to Sandy's landfall was studied, when winds blew water toward (fortunately!) the main inlet. How about computing and comparing setup for when the wind turned around and blew from the south after landfall, toward the nearly dead-end northern end of the bay? If we are interested in hazards, then this was the primary backbay damage-causing period of the event. I believe the local wind setup became quite large and abrupt at that time.

**The estimation of the wind setup during Sandy goes beyond the scope of the paper. The method can provide an estimate, but it is only a relative approximation. The available numerical model simulations of the area during Hurricane Sandy (Defne, Z., and Ganju, N.K., 2019, Collection of USGS Barnegat Bay hydrodynamic model simulations for Hurricane Sandy: U.S. Geological Survey data release, https://doi.org/10.5066/P99K85SW, available from the USGS Hurricane Sandy model portal: https://cmgdata.usgsportals.net/) represents a better estimate of the water level dynamics during the storm.**
**Nevertheless, we performed the estimation using the corrected implementation of the Wong & Moses-Hall formulation for both winds from the south (as requested) and also from the north. As the setup in the formulation is directly proportional to the wind stress, a wind stress of 1 Pa from the south would have resulted in elevations of around 2 m with the same pattern as in the new Figure 8 above. The formulation estimate for a wind stress of the same intensity (1 Pa) from the north is shown in the figure below. The figure shows the wind surge magnitude and provides an approximation for the gradient in water level caused by the wind stress. The total elevation would be the result of the addition of local wind setup plus offshore influence. So the elevation**

would match the offshore signal in the south near Little Egg Inlet and it would be a set-down (negative elevation with respect to MSL if no offshore elevation considered) in the north of the bay. Numerical model simulations (Defne and Ganju, 2019) show a similar pattern prior to Sandy's landfall. The magnitude of the wind effect (set-down with wind from the north, setup with winds from the south) is of the order of 1-2 m.

[Figure]

**The focus of the present work is not the study of the water level dynamics during Hurricane Sandy, but rather the characterization of the average response of the bay to hazards. The Sandy example in the paper is mostly used as a discussion point.**

During Sandy, water levels at Mantoloking rose about 2.5m in 12 hours when the wind rotated to come from the east and then south, reaching a maximum that was very close to the open ocean or Little Egg values (perhaps 30cm lower) (USGS station 01408168).
**The analysis of observations and model results during Hurricane Sandy goes beyond the scope of the current study. The analysis of numerical model results for Barnegat Bay during Hurricane Sandy is underway and the basic characteristics are available at the USGS model portal (https://cmgdata.usgsportals.net/).**

Does the analytical approach capture the effect of this large wind setup? If not, does this issue show that local wind setup can be a challenge for using the ADCIRC tide data to estimate storm hazards? Or, is Sandy too unusual of a case, in which case a nor'easter might be a better discussion point for the paper?
**The new corrected results of the wind setup effects show that the approach described in the paper represents a match with the expected behavior. The addition of the ADCIRC-derived offshore transfer and the analytical wind setup has the potential for adequately estimating water level in back-barrier bays.**

Using a back-of-the-envelope computation with an admittedly simpler, but well- established method to compute the wind setup for a 20m/s wind, 50km fetch, 2m deep backbay, if fully developed, is 2m (the Zuider-Zee equation, or similarly, making the computation using a steady state vertically averaged momentum budget - Pugh and Woodworth, 2014, p. 156 in Section 7.3 on Storm Surges). It takes only a matter of hours to fully develop. I agree on the 1Pa wind stress for Sandy- this is reasonable.

U=20m/s eg post-landfall Sandy
depth d=2m
fetch F=50000m
setup S= 0.000002 * F * U^2 /g /d = 0.000002 * 50000* 400/9.8/2 (Zuider-Zee equation) S = 2m setup
I believe "L" in the analytical wind formulation is the basin length (for each of several small basins). I am computing the wind setup for a 50km long backbay, so perhaps using a longer fetch. But I believe this is appropriate as they are really not disconnected and the ~40km of the northern half of the model domain is strongly connected (not divided up into separate bays).

**The new corrected estimates are consistent with the values described by the reviewer. The modified text in Section 6.1 of the paper now reads:**

**Ln 273-282: "The wind setup effect inside the bay due to local wind can also be estimated for Hurricane Sandy using the approach in Section 4.2. Maximum wind stress during the storm was about 1 Pa. To obtain a maximum effect (worst-case scenario) the wind was assumed to be persistently in the along-bay direction and that maximum stress was maintained for the duration of the storm. The maximum resulting water level considering the Wong and Moses-Hall method is linear with regard to wind stress magnitude (Figure 7b) and would have been 10 times larger than the setup in Figure 8b. The maximum wind setup would have been between 1 and 2 m, which was of the same order of magnitude as the surge produced from offshore sources. The cross-bay contribution to the wind setup during Sandy was comparatively small as wind direction was predominantly along-bay. Surge estimates from simple analytical formulations (State Committee for the Zuiderzee, 1926; Pugh, 1987) that do not consider storm duration produce similar magnitude results and are also dependent on the frictional response of the bay."**

Sandy's surge might be viewed as having a "slow" 1-m surge with timescale of 3 days, plus a fast 1-m surge with a timescale of 1 day. I think for either case the ADCIRC- based transfer results in this paper suggest reduced transfer (maybe 60%; figure 10) for Mantoloking. In contrast, Sandy, the worst extreme event, shows that local wind effects lead to a similar surge there as seen offshore (perhaps 90% transfer). The transfer uncertainty estimates in Figure 11 (eg 4% in the 2-day storm band?) aren't evaluating this wind setup contribution, so aren't worth much.

**The reviewer is correct about the relative magnitude of the two effects. The text of the paper now includes a section describing the need to add the wind effect to the ADCIRC derived offshore transfer:**

**Ln 314-319: "The effect of local wind setup will also need to be added to the ADCIRC-based estimate, especially during severe storms. The approach discussed in Section 5.2 or even a simpler surge calculation (e.g., from the steady state vertically averaged momentum equations, as in Pugh (1987), from the traditional report of the State Committee for the Zuiderzee (1926), or the updated frequency domain equivalent from Reef et al., 2018) could be used and the resulting elevation could be added to the offshore transfer estimate obtain based on the ADCIRC tides. Thus, the production of bay water level predictions will require accurate wind forecast products."**

I think this local wind effect pushing water into the northern end of the bay is what causes the unexplained high transfer at Mantoloking (Figures 4-5) in the 2-day storm band. The model captures it because it includes wind forcing (and Sandy), but a tide- only model will not.

**Local wind setup is likely the reason for the enhanced transfer response during 2012. The magnitude of the storm is not as important as the fact that during 2012 the general pattern of the pressure systems over the Atlantic was conducive to wind in the along-bay direction. The effect was a potential enhancement of the local wind setup that showed up in Mantoloking and not in other stations (e.g., Waretown) because the water was able to setup in the northern part of the bay. The text in Section 5.1 now includes a sentence describing this:**

**Ln 205-209: "The model reproduced the enhanced transfer in the storm band at Mantoloking during 2012, suggesting a physical mechanism for the change that the model was able to capture but remains unexplained. The likely explanation is that the location of the Azores-Bermuda high-pressure system over the Atlantic in 2012 (Mattingly et al., 2012), associated with the negative phase of the North Atlantic Oscillation, resulted in average winds that lined up with the axis of the Bay and caused enhanced wind setup in the northern part of the bay."**

To conclude, I think the method presented would, for storms, often be low-biased for peak water level risk estimates, due to local wind setup. I suggest proceeding very carefully and validating storm hazard estimates, using observational data.... or evaluating case studies carefully to determine whether local

wind setup corrections or larger uncertainties can be added for the storm driven flood hazards. Underestimating storm-driven flood risk is worse than not estimating it at all.

REFERENCE Pugh and Woodworth, 2014, Sea Level Science, Second Edition, Cambridge University Press, Cambridge, UK.

**We greatly appreciate the reviewer's comments that forced us to review the implementation of the wind setup approach discovering an error. When the error was corrected, the method of combining the transfer of offshore fluctuations, via the analytical or ADCIRC-based approaches, and the local wind setup effect showed the potential for estimating bay water levels in response to tidal and storm forcing. We agree that under or over-estimation of storm driven flood risk has severe consequences for coastal hazard mitigation. Proper skill assessment of the methodology will be needed before the implementation of this kind of approach for storm impact. The text now includes a discussion of this point:**

[revised manuscript text omitted]

Alfredo Lopez de Ar…, 5/31/2019 9:52 PM

Alfredo Lopez de Ar…, 2/20/2019 8:32 PM

Alfredo Lopez de Ar…, 5/31/2019 9:52 PM

Alfredo Lopez de Ar…, 2/20/2019 8:33 PM

Alfredo Lopez de Ar…, 5/31/2019 9:52 PM

Alfredo Lopez de Ar…, 2/20/2019 8:36 PM

Alfredo Lopez de Ar…, 2/20/2019 8:33 PM

Alfredo Lopez de Ar…, 5/31/2019 9:52 PM

Alfredo Lopez de Ar…, 5/31/2019 9:52 PM

Alfredo Lopez de Ar…, 2/20/2019 8:34 PM

Alfredo Lopez de Ar…, 5/31/2019 9:52 PM

Alfredo Lopez de Ar…, 2/20/2019 8:34 PM

Alfredo Lopez de Ar…, 5/31/2019 9:52 PM

Alfredo Lopez de Ar…, 2/20/2019 8:36 PM

Alfredo Lopez de Ar…, 2/20/2019 8:34 PM

**4.2 Local Wind Impact on Bay Model**

[revised manuscript text omitted]

$$\frac{\hat{\eta}_j}{\eta_o}(x,\omega) = 1 - \frac{\left(1-\frac{\eta_j}{\eta_o}(\omega)\right)\left(1-\frac{\eta\,\eta_{M_2}(x,\omega_{M_2})}{\eta\eta_{M_2}(offshore,\omega_{M_2})}\right)}{\left(1-\frac{\eta_j}{\eta_o}(\omega_{M_2})\right)} \qquad (13\,13)$$

where $\eta_j/\eta_o\,(\omega)$ is the transfer coefficient of the sub-embayment $i$ (single value) at frequency $\omega$, $\eta(x,\omega_{M_2})$ is the amplitude of the $M_2$ tidal fluctuations from the numerical model solution (spatially variable), and $\hat{\eta}_j/\eta_o\,(x,\omega)$ is the spatially variable adjusted transfer coefficient for sub-embayment $j$. The resulting adjusted transfer coefficients provide estimates of the spatial changes not only between adjacent sub-embayments but also inside each of the sub-embayments. The local wind effects on bay water level can be added to the impact from offshore fluctuations to obtain a combined local and remote water level response estimate.

**5 Results**

**5.1 Offshore transfer to bay**

[revised manuscript text omitted]

---

## Author Response (AR2)

**Response to Reviewer, comments in plain text, response in bold**
Anonymous Referee #3

Review of " Spatial distribution of water level impact to back-barrier bays " by Alfredo L. Aretxabaleta et. al.

The manuscript is addressed to a very relevant issue concerning the construction of a simple and fast running model aimed to provide sufficient accurate calculation of ensuing response (transfer) of the water level in a semi-enclosed bays, forced by offshore sea level fluctuations and local friction stress by winds acting on the surface of water body, modulated by bay geometry and bathymetry.
The authors propose a transfer estimates of offshore to bay water level using a combination of observations, analytical models based on appropriate simplifications of the bay system, and numerical simulations that provide, as underlined by the authors, the needed spatial distribution and more realistic frictional control.
The topic is interesting, methodology and results appear convincing and well documented.
Therefore, I think that the manuscript is worth to be accepted for publication in NHess.
**We thank the reviewer for the kind comments about the relevance of our paper.**

However, on a first reading, I found some difficulties for a sufficiently fast comprehension of some issues that instead appear, on a second reading, to be fundamental in the structure of the paper, in particular the issue related to the question how the remote forcing factors due to offshore water level modulations and local ones due to wind might be combined to provide a reliable, sufficiently accurate, representation of the water level trends within the bay.
I think this is one of the most relevant aspect of the proposal, that I suggest to the authors to revise accurately, before the final submission of the manuscript.
**We agree with the reviewer that this combination is critical for the proper representation of the water level response in bays.**

In the manuscript the issue is discussed, in rather simplified way, in paragraph 4.2. and 4.3 which refer to a very interesting paper of Wong and Moses-Hall (1998). These latter authors, on a base of a simple conceptual model with a very highly simplified bathymetry and geometry, derive the basic idea that the observed subtidal fluctuation in current and sea level within an estuary can be divided into two parts, linearly related, one forced by the local wind induced effect, and the other forced by the remote effect due to offshore sea level fluctuation.
Wong and Moses-Hall also observe that such uncoupling between the level modulations in the bay due to the two forcing factors, may not be readily transferable to a realistic estuary with complex bathymetry and geometry. In the latter case, wind driven flows might have a very complex structure with presence of a strong three-dimensional features characterized by a combination of vertical and horizontal recirculation flows. (In order to explore this aspect, the authors might refer to the following papers: HUNTER, J.R. and HEARN, C.J. (1987). "Lateral and Vertical Variation in Wind-Driven Circulation in Long, Shallow Lakes". J. Geophys. Res. 92, 13106-13114; Cioffi, Francesco, Francesco Gallerano, and Enrico Napoli. "Three-dimensional numerical simulation of wind-driven flows in closed channels and basins." Journal of Hydraulic Research 43.3 (2005): 290-301).
The dominance of horizontal or vertical recirculation flows depends mainly on the bay bathymetry, geometry and on the forcing factor characteristics (winds, tides, storm surge, etc…), amplitude and frequency. Depending on such characteristics the dynamics of the hydraulic systems might be linear, weakly, as well as, fully nonlinear with eventually possibility of resonance phenomena,
**The reviewer raises an important concern over the internal flow dynamics of estuaries and bays. The literature in this topic is quite extensive. The references mentioned have been added to the text. One important caveat is that in the current paper we are not explicitly pursuing a three-dimensional representation of the hydrodynamics in the bay, but rather we just focus on the water level response. The flow dynamics in bays are significantly more complex and three-dimensional and are quite nonlinear in many cases. The interaction between flow and bathymetry results in complex features that are highly variable in space and time. The water level on the other hand exhibits a much more linear response to forcing. Many studies, including Wong and Moses-Hall, have demonstrated the different responses of flow versus water level (e.g., Wong and Wilson, 1984; Garvine, 1985; Wong and DiLorenzo, 1988; Li and Valle-Levinson, 1999; Winant, 2004). We have modified the text to address the reviewer's concerns: Ln 156-159: "The local and remote effects can be combined in following the approach by Wong and Moses-Hall (1998). While bays can exhibit complex spatial responses to wind forcing**

especially in terms of currents (Csanady, 1973; Hunter and Hearn, 1987; Cioffi et al., 2005), the basic response can be summarized as the sum of local (wind) and remote (surge) forcings. The boundary condition for the local wind effect can be altered to account for the influence of offshore water level, $\eta_o$."

In the manuscript, the authors apply the proposed methodology, to a very complex water body with multiple mouths connecting the bay to the sea; in fact, their correctly use 3D simulations to identify the spatial distribution of the transfer response of the bay, and provide a very convincing criteria to correct the outputs of analytical model.

But, in the case of the simultaneal presence of wind and offshore level modulations the weakly nonlinearity of the system, in the specific case examined, should be demonstrated in such a way.

**We have modified the text to address the point raise by the reviewer.**

**Ln 179-183: "The local wind effects on bay water level can be added to the impact from offshore fluctuations to obtain a combined local and remote water level response estimate. In cases with simultaneous presence of wind and offshore level fluctuations, the system can respond in a weakly nonlinear manner and departures from the presented basic addition of the process are expected."**

The authors in the session in which results are discussed, try to demonstrate the correctness of this assumption in the specific case examined, but in my opinion not in a enough rigorous way. A reasonable effort to prove the validity of the assumption surely will increase the worth of the paper and its impact.

**The evaluation of the assumption will require a set of rather complex numerical simulations to isolate the distinct processes that interact to produce the complete water level signal. That experiment, while interesting, goes beyond the scope of the current study that focuses in a relatively simple methodology to describe water level response based on analytical formulations and easily available numerical model solutions.**

Just as a suggestion, I believe that the hypothesis of weakly nonlinearity ( derived by Wong and Moses-Hall ) in the specific case study examined, probably ( but this deserves a more detailed analysis) holds because the frequencies of the remote and local forcing factors are sufficient different. In particular, I suggest to the authors to revise paragraphs 4.2 and 4.3, and the related issues faced in the result discussion sessions.

**We have revised the document to address the reviewer's point of the complexity that including both local and remote effects can add. We have modified the text in several locations:**

[revised manuscript text omitted]

---

## Author Response (AR3)

**Response to Reviewer, comments in plain text, response in bold**
Anonymous Referee #4
* * *
Comments to the Author:
Dear Dr. Aretxabaleta et al,

I thank you for your new revised paper! It is much improved now. I should say, that there was one more late review on your paper, which came after I made my "minor revision" decision. It also requires minor revision and I kindly ask you to take these comments into account. Please, see the list below. As before I kindly ask you to submit your response to these comments and include, in the same author reply document, a track changes document between the old manuscript and the new one.

SUGGESTIONS FOR REVISION
The paper is much improved, and in good condition for publication. A few suggestions for minor edits are included below.

DETAILED INPUTS
All the paper's paragraphs have no spacing between them- I hope this isn't the way a final publication will look!
**Paragraph separation has been added, but the final formatting comes directly from Copernicus.**
Section 4.1: Consider adding text that explains why you define K (presumably to reduce the equations?) and what it represents (if simple to explain).
**We have added an explanation of the K expression:**
**Ln 150-152: "Assuming $\eta = \tilde{\eta}e^{i\omega t}$ and $u = \tilde{u}e^{i\omega t}$, where $\tilde{\eta}$ and $\tilde{u}$, represent the magnitude of the water level and velocity oscillations, respectively. Then, to reduce the size of the equations, we can define $K_n = \frac{h_n W_n g}{L_n\left(\frac{r_n}{h_n}+i\omega\right)}$ for $n$=1 ,…, 8, as the relative contribution of each sub-embayment based on its geometric and frictional characteristics."**

Section 4.2: This section generally seems better now, but I highly recommend some simple clarifications associated with application of Eq. 12 – after it is presented, add text clarifying how it is applied with k being an imaginary number, for those who are not specialists in derivations and applications of wave equations (a good part of the likely readership of the paper). I understand the imaginary wavenumber leads to exponential decay. But does one take the real part of the solution? The magnitude? I applied it and plotted the magnitude, and it seems to give similar results to yours.
**An explanation has been added to the text. It now reads:**
**Ln 156-160: "The wavenumber $k$ determines the spatial response of the transfer between the wind stress and the bay water level. The imaginary wavenumber part leads to exponential decay based on the frictional characteristics. The magnitude of water level is obtained from Equation 11, while the ratio of real to imaginary parts provides information about the phase lag between wind stress and water level."**

145 – If frequency is applied in radians, then "(angular)" frequency makes sense, but if frequency is applied in cycles (as I believe it is here), then I believe you should just use the term "cyclic frequency" or "frequency". So I think it's a bit confusing to say cyclic (angular) frequency. Choose what you feel is best, but give this some more thought.
**While I agree that it is important to be clear about the proper terminology, the main principle behind the expression does not change (only a factor is added). We have corrected the confusion and the term "angular" has been removed.**

line 160- should this have an equation number?
**It has an equation number as far as I can tell.**

Section 5.1
page 8 - this appears to be one long block of text (though difficult to tell due to no paragraph spacings) .... consider breaking up separate topics into topical paragraphs- eg the effects of the breach
**It had three paragraphs that now are more clearly separated.**

fig 7b- might explicitly state in the caption that x/L = 1 is north end of bay – just to ease interpretation
**Added to caption.**

line 323- "under developed" isn't a word pair?
**Corrected.**

I thank you very much for your submission and look forward to seeing the revised version of your manuscript!

[revised manuscript text omitted]
_1 (\widetilde{\eta_o}\phi_{LEI} - \widetilde{\eta_1})/L_1(i\omega + r_1/h_1) - h_2 W_2 (\widetilde{\eta_1} - \widetilde{\eta_2})/L_2(i\omega + r_2/h_2) \\ h_2 W_2 (\widetilde{\eta_1} - \widetilde{\eta_2})/L_2(i\omega + r_2/h_2) - h_3 W_3 (\widetilde{\eta_2} - \widetilde{\eta_3})/L_3(i\omega + r_3/h_3) \\ h_3 W_3 (\widetilde{\eta_2} - \widetilde{\eta_3})/L_3(i\omega + r_3/h_3) + h_4 W_4 (\widetilde{\eta_o}\phi_{BI} - \widetilde{\eta_3})/L_4(i\omega + r_4/h_4) - h_5 W_5 (\widetilde{\eta_3} - \widetilde{\eta_4})/L_5(i\omega + r_5/h_5) \\ h_5 W_5 (\widetilde{\eta_3} - \widetilde{\eta_4})/L_5(i\omega + r_5/h_5) - h_6 W_6 (\widetilde{\eta_4} - \widetilde{\eta_5})/L_6(i\omega + r_6/h_6) \\ h_6 W_6 (\widetilde{\eta_4} - \widetilde{\eta_5})/L_6(i\omega + r_6/h_6) + h_7 W_7 (\widetilde{\eta_o}\phi_{breach} - \widetilde{\eta_5})/L_7(i\omega + r_7/h_7) + h_8 W_8 (\widetilde{\eta_0}\phi_{PPC} - \widetilde{\eta_5})/L_8(i\omega + r_8/h_8) \
[revised manuscript text omitted]

Alfredo Lopez de Ar…, 7/25/2019 1:57 PM